# From Winning to Understanding: A Diagnostic Long-Horizon RTS Benchmark for LLMs

**Jiacheng Li** [*1] **Jiahui Liu** [*2] **Yuqing Wang** [*3] **Gaochen Cui** [4] **Xiao Zhang** [5]
**Qianchuan Zhao** [4] **Ziyou Zhang** [†4] **Chenghao Li** [†4]

## Abstract

Large language models (LLMs) are increasingly used as decision modules, yet existing benchmarks provide limited coverage of long-horizon, adversarial interaction while faithfully acting on human instructions. We introduce a long-horizon Red Alert RTS benchmark with a hierarchical interface in which LLMs output budgeted, low-frequency macro/tactical intents that are executed deterministically for standardized comparison. The benchmark evaluates (i) robustness to "rules-as-variable" perturbations via rule-style shifts, (ii) competitive strength via Elo-style ratings from head-to-head matches, and (iii) human steerability via standardized language interventions. Beyond win/loss, we log economy growth/spending, combat loss ratio, and visibility coverage to diagnose long-horizon failure modes. Overall, the benchmark provides a reproducible and diagnostic testbed for robustness and controllability in long-horizon adversarial decision making. Experiments on six frontier LLMs reveal pronounced capability fragmentation and matchup asymmetries, demonstrating that single aggregate scores obscure opponent- and instruction-dependent failure modes.

## 1. Introduction

Large language models (LLMs) are increasingly deployed as decision modules that plan, reason, and invoke tools under closed-loop feedback, rather than as pure text generators (Yao et al., 2023; Schick et al., 2023; Ahn et al., 2022; Wang et al., 2023; Shinn et al., 2023). Correspondingly, evaluation has evolved along two largely complementary tracks. Static benchmarks such as MMLU (Hendrycks et al., 2021), GSM8K (Cobbe et al., 2021), and HumanEval (Chen et al., 2021) offer clean, reproducible measurements of knowledge, reasoning, and code correctness. In parallel, interactive "LLM-as-agent" benchmarks assess multi-step instruction following and tool-mediated task completion in web/software environments (Liu et al., 2024; Zhou et al., 2024; Deng et al., 2023; Xie et al., 2024; Drouin et al., 2024) and grounded interactive worlds (Yao et al., 2022; Shridhar et al., 2021; Wang et al., 2022; Fan et al., 2022; Liu et al., 2018).

Together, these tracks have clarified many facets of LLM competence; however, as LLMs move from one-off problem solving to embedded control loops, the key question shifts from single-shot correctness to interactive reliability: can a model maintain goal-directed behavior across many rounds of observation, action, and feedback, under partial information and delayed consequences? Existing interactive benchmarks begin to address this shift, yet they largely focus on non-adversarial settings and short- to medium-horizon objectives, and thus only weakly probe failure modes driven by strategic commitment and cross-stage consequences—where early decisions constrain, amplify, or even preclude later options. Moreover, they provide limited leverage to test instruction responsiveness in the way human operators actually intervene, leaving it unclear whether agents can faithfully instantiate and sustain tactical directives over time in realistic human–AI interaction.

To address this gap, we introduce a long-horizon diagnostic benchmark centered on real-time strategy (RTS) competition in *Red Alert*, where two players start with an initial base and the objective is to eliminate all enemy combat units and buildings within a 30-minute time limit. A match ends in a win upon complete enemy elimination, a loss upon total friendly unit destruction, or a draw if neither occurs within the time limit. Building on OpenCodeAlert (OpenCodeAlert, 2025), we standardize evaluation with an explicit decision budget and a hierarchical interface: the LLM pro-

---

[*]Equal contribution [†]Equal advising [1]University of Chinese Academy of Sciences, Beijing, China [2]Shenyang University of Technology, Shenyang, China [3]Stony Brook University, Stony Brook, NY, USA [4]Department of Automation, Tsinghua University, Beijing, China [5]University of Science and Technology Beijing, Beijing, China. Correspondence to: Ziyou Zhang <zbw6596@163.com>, Chenghao Li <lch.tsinghua@gmail.com>.

*Proceedings of the 43rd International Conference on Machine Learning*, Seoul, South Korea. PMLR 306, 2026. Copyright 2026 by the author(s).

duces budgeted, low-frequency macro/tactical intents, while a deterministic executor translates intents into low-level commands. This design preserves the long-horizon adversarial structure while decoupling high-level decision making from low-level execution efficiency, enabling fairer comparisons and clearer diagnosis. We position this benchmark not as a universal proxy for all long-horizon planning, but as a diagnostic testbed for recurring core properties: partial observability, delayed consequences, adaptive opponents, and sustained responsiveness to human interventions.

Within this framework, we evaluate LLM agents along three complementary axes. (1) Robustness to rules-as-variable perturbations: controlled rule injections and constraint shifts that test whether agents remain effective while complying with modified requirements. (2) Competitive strength: head-to-head performance summarized by Elo-style ratings over repeated matches. (3) Human steerability: natural-language interventions injected at predefined times or state triggers to test whether agents make behaviorally grounded adjustments mid-trajectory, rather than merely acknowledging instructions in text. Beyond win/loss, we log lightweight process signals—including economy growth and spending, combat exchange efficiency, and exploration coverage—to support attribution of long-horizon failure.

Overall, our experiments demonstrate that long-horizon strategic decision quality cannot be reliably summarized by a single number. Performance varies substantially across opponent regimes and behavioral objectives: models that appear strong against fixed, style-controlled baselines may not dominate in peer-adaptive cross-play, and agents that excel on isolated tactical skills can still struggle to translate those strengths into consistent competitive outcomes. These gaps motivate future LLM decision-making research toward agents that are simultaneously robust to style shifts, resilient under adaptive competition, and reliably steerable under human intent over long horizons.

## 2. Related Work

### 2.1. From Static Competence to Interactive Agent Evaluation

A large body of benchmark work evaluates LLMs in static settings, providing reproducible measurements of knowledge and reasoning. Representative suites include MMLU (Hendrycks et al., 2021) for broad factual and conceptual understanding, GSM8K (Cobbe et al., 2021) for math word problems, and HumanEval (Chen et al., 2021) for code generation. BIG-bench (Srivastava et al., 2022) and HELM (Liang et al., 2022) broaden coverage across tasks, domains, and evaluation dimensions. These benchmarks are indispensable for tracking core capabilities, but they deliberately abstract away interaction dynamics—state evolution, de-

layed feedback, and error accumulation—that define long-horizon decision making.

To bridge this gap, recent work increasingly treats LLMs as interactive agents, evaluating multi-step behavior in environments that require planning and tool use. AgentBench (Liu et al., 2024) provides a multi-environment evaluation suite for LLM-as-agent performance. Web-based benchmarks such as WebArena (Zhou et al., 2024) and datasets such as Mind2Web (Deng et al., 2023) emphasize realistic web interaction, while OSWorld (Xie et al., 2024) and WorkArena (Drouin et al., 2024) push toward computer- and enterprise-style tasks with execution-based evaluation. Complementary efforts test tool invocation and API orchestration more directly (Yao et al., 2022), reflecting the growing role of external tools in agentic systems. Collectively, these benchmarks improve realism and reduce evaluation ambiguity, but many tasks are still dominated by short- to medium-horizon trajectories, and they often provide limited leverage to study adversarial adaptation or strong path dependence.

### 2.2. Controllability, Constraints, and Human Steerability

A central deployment requirement for decision-making agents is not only to succeed, but to do so under constraints and in alignment with user intent. Alignment and instruction-following research (e.g., RLHF-style tuning) aims to make model behavior more responsive to human preferences and constraints (Ouyang et al., 2022), while related lines explore rule- or principle-based control (Bai et al., 2022). In evaluation, Elo-style model comparisons and judge-based protocols have been used to summarize relative capability and preference under head-to-head settings (Elo, 1978; Zheng et al., 2023). At the systems level, long-horizon interaction has been studied through simulacra of persistent agent behavior (Park et al., 2023), while many agent frameworks incorporate memory and reflection to stabilize behavior over multi-step trajectories (Shinn et al., 2023), and memory management has been studied explicitly to extend effective horizons under finite context (Packer et al., 2023).

Most existing benchmarks, however, operationalize controllability only implicitly—typically as end-task success given a natural-language instruction—and offer limited instrumentation to verify whether an agent faithfully instantiates structured intent over time. Our benchmark instead treats rules-as-variable and human suggestions as controlled interventions, and pairs them with log-computable, process-level measurements (e.g., deviation and persistence under interventions, along with tempo/efficiency signals) that disentangle robust win conversion from mere non-losing behavior. This is particularly important in long-horizon adversarial settings, where apparent compliance can be achieved by safe,

draw-prone play, yet sustained alignment requires stable execution and timely commitment under shifting constraints and adaptive opponents.

# 3. Benchmark Detail

To obtain an evaluation that is both diagnostic and practically relevant, we organize our benchmark into three complementary tracks: (i) *LLM vs. rule-based AI*, which enables controlled attribution against a stationary opponent set comprising four *style-distinct* rule-based agents; (ii) *LLM vs. LLM*, which yields opponent-agnostic relative rankings via Elo-style ratings; and (iii) *human command following*, which quantifies instruction grounding and coordination quality beyond competitive strength.

## 3.1. Common Setup

All experiments are conducted in the OpenRA real-time strategy environment using the open-source Red Alert implementation.[1] We use the default OpenRA ruleset without modification: unit attributes, combat mechanics, and economic systems remain unchanged. All evaluations are run on a single canonical map with a fixed initial configuration. We adopt the engine's native elimination win condition, under which the objective is to eliminate all enemy combat units and buildings within a 30-minute time limit.

As shown in Fig. 1, agents interact with the environment via a closed-loop decision process. At each decision step, the LLM generates a batched, multi-action intent with a strict per-round action budget. A unified deterministic executor translates these intents into low-level engine commands and enforces action legality; the next LLM decision round begins only after all batched actions from the current round are fully completed. This design decouples high-level strategic reasoning from low-level execution efficiency (APM), ensuring the benchmark measures reasoning depth rather than micro-operation speed. Response latency—the interval between action completion and the next decision round—is retained as an intrinsic capability characteristic rather than controlled away.

A typical 30-minute match comprises 28–38 high-level decision rounds, with agents deciding every 47–64 seconds (approximately 0.93–1.27 decisions per minute). At each step, the LLM outputs a normalized, structured JSON action list consisting of unit-level commands. The action space includes building commands for constructing structures and unit commands (move, attack, and attack move). The attack move command allows units to engage enemy combat units encountered en route, but does not automatically target enemy buildings, which require explicit attack commands.

All actions are executed by a deterministic executor that enforces action legality and translates valid commands into low-level engine instructions, without introducing additional decision logic. Each match is initialized independently: no experience, memory, or internal state is carried over across games.

For transparency and reproducibility, all models are queried via their official native APIs with default decoding settings and evaluated under strictly zero-shot conditions; no few-shot examples of the JSON action space are provided in the prompts. The system prompt for Tracks A and B is provided in Appendix G. The prompt for Track C (differing in objective clauses) and full interaction logs are available online.[2]

## 3.2. Track A: LLM vs. Rule-based AI

Track A evaluates an LLM agent $\pi$ against a predefined suite of rule-based opponents $\{\pi_{\text{rule}}^{(k)}\}_{k=1}^{K}$ covering diverse strategic styles. Specifically, Turtle AI focuses on defense and slow buildup; Normal AI follows a balanced strategy; Rush AI emphasizes early attacks; and Naval AI focuses on naval control and water-based attacks. To make performance differences interpretable beyond the terminal outcome, we report win rate together with a small set of process-level diagnostics that reflect the RTS capability pipeline: scouting, economic throughput/conversion, and combat exchange.

**Scouting.** Let $\text{VisCells}(t)$ be the set of map cells visible to $\pi$ at time $t$. We quantify exploration as

$$\text{Scouting} = \left| \bigcup_{t \leq T} \text{VisCells}(t) \right|. \qquad (1)$$

Low Scouting typically indicates delayed threat detection and poor strategic branching due to partial observability.

**Economy.** Let $\Delta\text{In}(t)$ and $\Delta\text{Out}(t)$ denote resource inflow and outflow at time $t$. We measure the *per-minute* income rates,

$$\text{MC} = \frac{1}{T} \sum_{t \leq T} \Delta\text{In}(t), \qquad (2)$$

to capture earning efficiency. In addition, we compute an overall expenditure ratio that accounts for initial money $M_0$,

$$\text{MU} = \frac{\sum_{t \leq T} \Delta\text{Out}(t)}{\sum_{t \leq T} \Delta\text{In}(t) + M_0}. \qquad (3)$$

**Combat exchange.** To assess trade efficiency, let $\text{LossCost}_{\text{self}}$ and $\text{LossCost}_{\text{enemy}}$ be the total destroyed

---

[1]https://www.openra.net/; https://github.com/OpenRA/OpenRA

[2]https://github.com/lich14/llm_fight

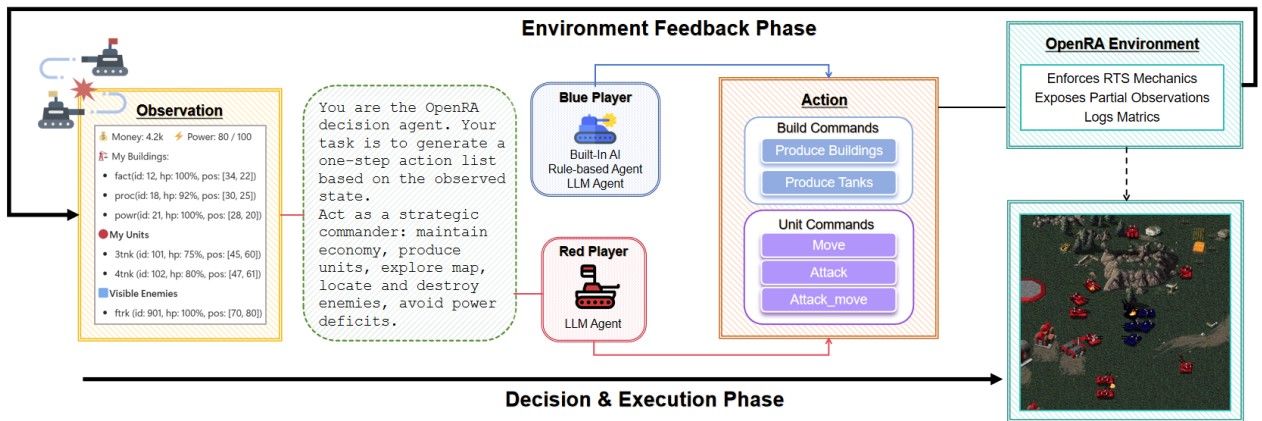

*Figure 1.* Overview of the LLM–OpenRA benchmark. LLM agents receive structured, partially observable observations from the environment, including resources, units, buildings, and visible enemies. Each agent generates a one-step action list, containing building and unit commands that execute tactical objectives such as pincer attacks, scouting, or coordinated assaults. The environment executes these actions, enforces RTS rules, updates the world state, and logs metrics including economy, combat outcomes, and unit losses for systematic evaluation.

unit/building costs (or the closest log-supported proxy). We define a cost-based kill–loss ratio:

$$\text{KLR} = \frac{\text{LossCost}_{\text{enemy}} + \epsilon}{\text{LossCost}_{\text{self}} + \epsilon}. \qquad (4)$$

Values $\text{KLR} > 1$ indicate favorable exchanges, while $\text{KLR} < 1$ suggests inefficient fights or poor target selection/positioning.

Together, Scouting, MC, MU, and KLR explain why each LLM agent wins or loses, disentangling scouting failures from economic bottlenecks and combat inefficiency, rather than conflating them into the terminal win/loss signal.

### 3.3. Track B: LLM vs. LLM

Track B focuses on evaluating a fixed set of LLM agents $\Pi = \{\pi_1, \ldots, \pi_N\}$ via pairwise matches to produce a relative strength leaderboard. Unlike Track A, which can be sensitive to the choice of rule-based opponents, Track B uses cross-play among diverse LLMs to produce a robust, opponent-agnostic leaderboard signal. We adopt the Davidson extension of the Bradley–Terry paired-comparison model with ties. This model is compatible with the standard Elo rating scale and supports incomplete schedules as well as order-invariant estimation.

For each pair $\{i, j\}$ of agents, we collect the counts of results $(w_{ij}, w_{ji}, t_{ij})$, where $w_{ij}$ denotes the number of wins of $\pi_i$ over $\pi_j$, and $t_{ij}$ the number of draws. The Davidson model assigns each agent $\pi_i$ a positive strength parameter $\alpha_i > 0$ and a global tie propensity $\delta \geq 0$. Under this parameterization, the probabilities of a win, draw, or loss for $\pi_i$ against $\pi_j$ are given by

$$(p_{ij}^{\text{win}}, p_{ij}^{\text{loss}}, p_{ij}^{\text{draw}}) = \frac{(\alpha_i, \alpha_j, \delta\sqrt{\alpha_i\alpha_j})}{\alpha_i + \alpha_j + \delta\sqrt{\alpha_i\alpha_j}}, \qquad (5)$$

We estimate $\{\alpha_i\}_{i=1}^N$ and $\delta$ by maximizing the joint log-likelihood of all observed outcomes,

$$\mathcal{L}(\boldsymbol{\alpha}, \delta) = \sum_{1 \leq i < j \leq N} \Big( w_{ij} \log p_{ij}^{\text{win}} + \\ w_{ji} \log p_{ij}^{\text{loss}} + t_{ij} \log p_{ij}^{\text{draw}} \Big), \qquad (6)$$

subject to an identifiability constraint ($\sum_i \log \alpha_i = 0$). To facilitate interpretation and comparison with existing benchmarks, we convert $\alpha_i$ into an Elo-style scalar rating using a base-10 logarithmic transformation:

$$\rho_i = 400 \log_{10}(\alpha_i) + 1500, \qquad (7)$$

where the offset 1500 sets the reference point of the rating scale. In addition, we release the pairwise win-rate matrix $\{W_{ij}\}$ to expose matchup-specific strengths and weaknesses and possible non-transitivities, which can be obscured by a single scalar rating.

### 3.4. Track C: Human Command Following

Track C evaluates controllability under human intent: whether an agent faithfully grounds high-level tactical instructions into coordinated multi-unit behavior over time, rather than merely securing a favorable outcome. The manually designed opponent uses automatic firing, leaving the LLM solely responsible for tactical positioning and execution. This complements Tracks A and B, which quantify competitive competence but cannot capture whether an

agent acts as a reliable commander that translates directives into coherent multi-unit action; an agent may win while ignoring commands, or follow them in degenerate ways (e.g., suicidal compliance). Track C therefore scores realized behaviors via log-computable geometric and role-based metrics, turning qualitative commands into reproducible, comparable quantitative measures.

**C.1 Encirclement.** In this tactical scenario, we evaluate LLMs' ability to execute a classic pincer attack maneuver. The red team is instructed to surround the blue team's tank formation from multiple directions while maintaining optimal attack distance. Success requires coordinating multiple units to form an even distribution around the target, balancing proximity for effective firepower with tactical positioning to avoid clustering. This tests the LLM's spatial reasoning and multi-unit coordination capabilities.

Let the $i$-th red unit be at $p_i(t) = (x_i(t), y_i(t)) \in \mathbb{R}^2$ at time $t$, and an enemy location be at $q = (x(t), y(t)) \in \mathbb{R}^2$. Define its target distance and bearing angle as

$$
\begin{aligned}
d_i(t) &= \|p_i(t) - q\|_2, \\
\theta_i(t) &= \operatorname{atan2}(y_i(t) - y, x_i(t) - x) \in (-\pi, \pi].
\end{aligned}
\tag{8}
$$

Let $d_{\min}$ denote the firing radius. We use the standard positive-part operator $[z]^+ = \max(z, 0)$.

**Participation term.** We quantify how many units are effectively participating via a smooth penalty on being outside the engagement radius, with $\alpha > 0$:

$$
\operatorname{Part}(t) = \frac{1}{n} \sum_{i=1}^{n} \exp\left(-\alpha \left[d_i(t) - d_{\min}\right]^+\right).
\tag{9}
$$

**Coverage term.** Let $\mathcal{A}(t) \triangleq \{i \in \{1, \dots, n\} : d_i(t) \leq d_{\min}\}$ be the set of actively firing units, and let $m(t) = |\mathcal{A}(t)|$. Sort the active angles $\{\theta_i(t)\}_{i \in \mathcal{A}(t)}$ into $-\pi < \phi_1(t) \leq \cdots \leq \phi_m(t) \leq \pi$ (with $m = m(t)$), and define the cyclic angular gaps

$$
g_j(t) = \begin{cases} \phi_{j+1}(t) - \phi_j(t), & j \leq m - 1, \\ 2\pi - \left(\phi_m(t) - \phi_1(t)\right), & j = m. \end{cases}
\tag{10}
$$

We measure clustering by the squared $\ell_2$ energy of the gap vector, $\sum_{j=1}^{m} g_j(t)^2$, which is minimized when the gaps are equal. We define the instantaneous coverage score as

$$
\operatorname{Cov}(t) = \exp\left(-\beta \sum_{j=1}^{m(t)} g_j(t)^2\right) \cdot \exp\left(\frac{4\pi^2}{n}\beta\right),
\tag{11}
$$

where $\beta > 0$ controls sensitivity to angular non-uniformity.

**Instantaneous encirclement quality.** Combining the participation and coverage terms yields the instantaneous encirclement quality $\operatorname{Enc}(t) = \operatorname{Part}(t) \cdot \operatorname{Cov}(t)$. For benchmarking, we report the time-average of $\operatorname{Enc}(t)$ over the evaluation window from the first firing time $t_f$ to the target elimination time $t_e$, denoting as $\operatorname{Score}_{\operatorname{Enc}}$.

**C.2 Front–Back Assault.** This scenario evaluates an LLM's ability to coordinate heterogeneous units under adversarial target selection, emphasizing value-aware formation control, risk-sensitive resource allocation, and multi-agent positioning to sustain effective team firepower. The adversary prioritizes the nearest unit, so the LLM must implicitly learn to place more expendable units in the front while preserving high-value units in the back, all while maintaining engagement conditions so that the team can continuously apply damage. Overall, this tests strategic reasoning about unit prioritization and spatial ordering in a dynamic multi-agent engagement.

**Participation term.** We reuse the participation (distance) term $\operatorname{Part}(t)$ from Eq. (9) to downweight units outside the engagement radius.

**Radial ordering term.** Each red unit has attack power $a_i$ and health $h_i$, and the key requirement is that units with larger value ratio $\rho_i = a_i/h_i$ should be placed at larger $d_i(t)$. We operationalize this as a *weighted discordant-pair* penalty between the two orderings induced by $\{d_i(t)\}$ and $\{\rho_i\}$. Define per-unit engagement weights

$$
w_i(t) = \exp\left(-\alpha \left[d_i(t) - d_{\min}\right]^+\right).
\tag{12}
$$

Consider all unordered pairs $(i, j)$ with $i < j$. A pair is *discordant* if its radial order disagrees with its $\rho$-order. We assign larger penalties to more consequential mistakes by weighting each discordant pair by both engagement and value disparity. The instantaneous discordance mass is

$$
\operatorname{Disc}(t) = \sum_{1 \leq i < j \leq n} \mathbf{1}[(d_i(t) - d_j(t))(\rho_i - \rho_j) < 0] \\ w_i(t)w_j(t)\left|\rho_i - \rho_j\right|,
\tag{13}
$$

and we normalize by the maximum attainable pair mass

$$
\operatorname{Norm}(t) = \sum_{1 \leq i < j \leq n} w_i(t)w_j(t)\left|\rho_i - \rho_j\right|.
\tag{14}
$$

The radial ordering score is

$$
\operatorname{Ord}(t) = 1 - \frac{\operatorname{Disc}(t)}{\operatorname{Norm}(t) + \varepsilon},
\tag{15}
$$

where $\varepsilon > 0$ is a small constant for numerical stability.

**Instantaneous front–back formation quality.** We define the instantaneous front–back quality as $\operatorname{FB}(t) = \operatorname{Part}(t) \cdot \operatorname{Ord}(t)$.

We denote the time-average of $\operatorname{FB}(t)$ over the evaluation window from $t_f$ to $t_e$ as $\operatorname{Score}_{\operatorname{FB}}$.

**C.3 Visibility Coverage.** This scenario evaluates an LLM's ability to execute reconnaissance operations by maximizing map exploration coverage. Rather than optimizing immediate combat outcomes, the goal is to assess *situational awareness* and *information gathering*: effective policies should spread units to expand the team's collective field of view, avoid redundant observation, and sustain coverage over a time window. This tests strategic planning, spatial coverage optimization, and resource allocation under uncertainty.

We assume the environment is discretized into a finite set of grid cells, denoted by MapCells. Given an anchor time $t_c$ and a horizon length $\Delta > 0$, we define the *exploration rate* as the fraction of map cells that are observed at least once during the window $[t_c, t_c + \Delta]$:

$$\text{Score}_{\text{VC}} = \frac{\left| \bigcup_{t \in [t_c, t_c + \Delta]} \text{VisCells}(t) \right|}{|\text{MapCells}|}. \quad (16)$$

## 4. Experiments

Our experimental protocol follows Sec. 3. We instantiate the three-track evaluation: (i) *LLM vs. rule-based AI*, (ii) *LLM vs. LLM*, and (iii) *human command following*, and evaluate all models under identical interfaces, decision budgets, and aggregation rules.

We evaluate DeepSeek-V3.2, Qwen-3-Max, Gemini-2.5-Flash, GPT-4o, Claude-Sonnet-4.5, and Kimi-K2.5-FW, which are widely used LLMs spanning both open-weight and proprietary families. For brevity, we henceforth refer to these models as DeepSeek, Qwen, Gemini, GPT-4o, Claude, and Kimi, respectively. Specifically, DeepSeek (DeepSeek-AI, 2025) and Qwen (Yang et al., 2025; Qwen Team, 2025) serve as strong open-weight baselines for reproducible research, while Gemini (Google, 2025), GPT-4o (OpenAI, 2024), Claude (Anthropic, 2025), and Kimi (Moonshot AI, 2025) are widely deployed proprietary models with competitive agentic reasoning capabilities. Unless otherwise stated, all models are queried via official APIs with default decoding and are subjected to the same observation/action interface.

### 4.1. Track A: LLM vs. Rule-based Opponents

In Track A, we repeat each matchup five times and report the mean across runs in Table 1 following Sec. 3. Frequent draws reflect low-aggression matchups between conservative agents that stalemate before either side achieves elimination. Decisive games typically conclude within 20 minutes, so the 30-minute bound primarily truncates non-aggressive encounters that would otherwise persist indefinitely.

Performance is strongly opponent-conditioned, and models separate most clearly by whether they can produce *wins* across multiple scripted styles. GPT-4o is the only agent that remains loss-free while securing wins against multiple opponent archetypes. Claude achieves the highest overall win rate and also converts wins across multiple styles, though with greater opponent-dependent variability. Qwen and Gemini are largely draw-dominant: they occasionally secure wins against more open play styles but frequently converge to stalemate-like outcomes against the most defensive opponents. In contrast, DeepSeek never achieves a win and incurs losses under several opponent styles, while Kimi also records multiple losses and shows a relatively low win rate, suggesting difficulties in adapting to more aggressive scripted strategies.

Beyond win/draw/loss, the macro signals reveal a clear decoupling between economic throughput and combat effectiveness. Claude achieves the highest kill-loss ratio and military utility, indicating aggressive resource conversion into favorable exchanges, yet its mineral throughput remains moderate. GPT-4o, by contrast, sustains the highest mineral throughput across all styles, suggesting superior economic scaling that does not always translate into proportionally higher win rates due to draw-prone play. Kimi shows the weakest overall macro profile, with sharp degradation against aggressive opponents, aligning with its heavy losses in those matchups. DeepSeek exhibits clear instability: its mineral throughput drops sharply against Normal AI and remains substantially lower than GPT-4o under Rush and Turtle, consistent with its elevated losses in those matchups.

The remaining diagnostics confirm that single-factor strength is insufficient. For example, Gemini achieves a very high KLR against Naval AI yet records only draws, while Claude pairs a similarly high KLR with decisive wins. Conversely, Kimi converts a moderate Naval KLR into a solid win, yet its KLR collapses against Rush where its economy also crumbles—showing that combat advantage without economic resilience is fragile. Overall, victory arises from coordinated performance across combat, economy, and information dimensions rather than dominance in any single metric (Appendix B). A radar-chart summary of cross-style capability profiles is provided in Appendix A (Fig. 16).

### 4.2. Track B: LLM vs. LLM

Fig. 2 shows the pairwise cross-play win-rate matrix. For each ordered model pair, we run five head-to-head matches and compute the average score using the standard scheme (win = 1, draw = 0.5, loss = 0).

The matrix reveals a pronounced hierarchy with strong matchup asymmetries. Claude dominates all other models, achieving perfect or near-perfect win rates against Gemini, Kimi, Qwen, and DeepSeek, while holding a narrower but consistent edge over GPT-4o. Gemini occupies a clear sec-

| Model | Opponent | Win | Draw | Loss | KLR | MC | MU | Scouting |
|-------|----------|-----|------|------|-----|-----|-----|----------|
| Claude | Naval AI | 1.00 | 0.00 | 0.00 | 22.12 | 2637.7 | 0.882 | 449.2 |
| | Normal AI | 1.00 | 0.00 | 0.00 | 3.93 | 3007.4 | 0.884 | 423.0 |
| | Rush AI | 0.20 | 0.40 | 0.40 | 1.23 | 3706.2 | 0.809 | 303.8 |
| | Turtle AI | 0.60 | 0.40 | 0.00 | 4.29 | 2695.7 | 0.833 | 425.0 |
| | *Mean across opponents* | **0.70** | 0.20 | 0.10 | **7.89** | 3011.7 | **0.852** | **400.2** |
| DeepSeek | Naval AI | 0.00 | 1.00 | 0.00 | 5.34 | 4350.9 | 0.566 | 355.6 |
| | Normal AI | 0.00 | 0.80 | 0.20 | 1.43 | 2717.0 | 0.712 | 295.4 |
| | Rush AI | 0.00 | 0.60 | 0.40 | 1.02 | 3162.1 | 0.727 | 263.4 |
| | Turtle AI | 0.00 | 0.60 | 0.40 | 1.66 | 3195.6 | 0.692 | 261.8 |
| | *Mean across opponents* | 0.00 | 0.75 | 0.25 | 2.36 | 3356.4 | 0.674 | 294.1 |
| Kimi | Naval AI | 0.60 | 0.40 | 0.00 | 12.49 | 2797.9 | 0.660 | 349.8 |
| | Normal AI | 0.00 | 0.60 | 0.40 | 0.85 | 2004.2 | 0.607 | 121.4 |
| | Rush AI | 0.00 | 0.20 | 0.80 | 0.39 | 1714.0 | 0.530 | 70.0 |
| | Turtle AI | 0.00 | 0.60 | 0.40 | 1.28 | 2406.2 | 0.695 | 235.4 |
| | *Mean across opponents* | 0.15 | 0.45 | 0.40 | 3.75 | 2230.6 | 0.623 | 194.2 |
| Qwen | Naval AI | 0.00 | 1.00 | 0.00 | 9.49 | 4552.8 | 0.603 | 400.6 |
| | Normal AI | 0.20 | 0.80 | 0.00 | 1.49 | 3048.2 | 0.731 | 385.0 |
| | Rush AI | 0.20 | 0.80 | 0.00 | 1.81 | 2856.3 | 0.777 | 306.2 |
| | Turtle AI | 0.00 | 1.00 | 0.00 | 2.45 | 3492.0 | 0.742 | 456.8 |
| | *Mean across opponents* | 0.10 | 0.90 | 0.00 | 3.81 | 3487.4 | 0.713 | 387.1 |
| Gemini | Naval AI | 0.00 | 1.00 | 0.00 | 18.70 | 3128.4 | 0.748 | 330.8 |
| | Normal AI | 0.40 | 0.60 | 0.00 | 3.47 | 2066.5 | 0.794 | 263.0 |
| | Rush AI | 0.20 | 0.80 | 0.00 | 1.74 | 2910.6 | 0.853 | 266.6 |
| | Turtle AI | 0.00 | 1.00 | 0.00 | 2.61 | 2986.6 | 0.764 | 290.4 |
| | *Mean across opponents* | 0.15 | 0.85 | 0.00 | 6.63 | 2773.0 | 0.790 | 287.7 |
| GPT-4o | Naval AI | 0.80 | 0.20 | 0.00 | 19.49 | 4885.2 | 0.784 | 342.0 |
| | Normal AI | 0.20 | 0.80 | 0.00 | 2.48 | 4242.7 | 0.882 | 348.2 |
| | Rush AI | 0.20 | 0.80 | 0.00 | 2.85 | 4545.4 | 0.859 | 354.4 |
| | Turtle AI | 0.60 | 0.40 | 0.00 | 3.62 | 5093.1 | 0.804 | 368.6 |
| | *Mean across opponents* | 0.45 | 0.55 | 0.00 | 5.74 | **4691.6** | 0.832 | 353.3 |

*Table 1.* Track A results: performance of LLM agents against rule-based AI opponents (each match lasts up to 30 minutes). Important average metrics are bolded to highlight key comparative strengths.

ond tier, consistently defeating the lower four models with strong margins, though it records no wins against Claude. Kimi and GPT-4o form a closely matched mid-tier: their direct matchup ends in a draw, and both dominate Qwen and DeepSeek while losing to Claude. Notably, Kimi holds a draw against Gemini where GPT-4o suffers a heavy loss, whereas GPT-4o fares better against Claude than Kimi does. Qwen and DeepSeek trail substantially and form the bottom tier, with Qwen holding a slight edge via a draw in their direct matchup and marginally better outcomes against all other opponents.

Based on the Davidson extension of the Bradley–Terry paired-comparison model described in Sec. 3.3, we compute Elo-style ratings in Table 2. Claude ranks first by a clear margin, followed by Gemini. Kimi and GPT-4o occupy a closely contested third–fourth tier, while Qwen and DeepSeek form a weaker fifth–sixth tier. The Track B ranking differs sharply from Track A: GPT-4o leads Track A in macro throughput and win conversion against scripted opponents, yet falls to the mid-tier in adaptive cross-play, while Claude—which showed strong but not dominant Track A performance—rises to the top once facing peer opponents. Conversely, Gemini retains a top-tier position across both tracks, and Kimi—which struggled in Track A—proves more competitive in head-to-head settings than its rule-based record would suggest.

To verify ranking stability under limited samples, we repeated the Rush AI evaluation 15 times and conducted a pair-stratified bootstrap analysis (1000 replicates); results confirm that the top-two ordering is highly stable, while the Qwen–DeepSeek separation remains comparatively more uncertain (Appendix D).

### 4.3. Track C: Human Command Following

Table 3 reports the performance of six frontier LLMs on three tactical benchmarks. The metrics target complementary capabilities: encirclement quality captures spatial coordination for pincer-style maneuvers, front–back formation

*Table 2.* Overall cross-play score.

| Model | Score |
| --- | --- |
| Claude | **2136.02** |
| Gemini | 1630.38 |
| GPT-4o | 1602.90 |
| Kimi | 1548.37 |
| Qwen | 1160.07 |
| DeepSeek | 1036.53 |

quality measures role-consistent positioning under heterogeneous unit attributes, and visibility coverage quantifies the efficiency of reconnaissance coverage. We repeat each match five times and report the mean and standard deviation across runs. To make the comparison more intuitive, we include visual summaries of how the six LLM agents respond to the three human command intents in Appendix A.

**Encirclement.** Claude, GPT-4o, and DeepSeek form a top tier on this symmetric spatial coordination task, all achieving encirclement scores above 0.60 with relatively low variance. Claude leads narrowly, suggesting particularly stable geometric reasoning. Qwen occupies a middle tier, while Kimi and Gemini fall behind, with Gemini recording the weakest performance despite its strength on other tactical dimensions. This pattern indicates that homogeneous unit coordination around a single target is not uniformly mastered even by strong generalist models.

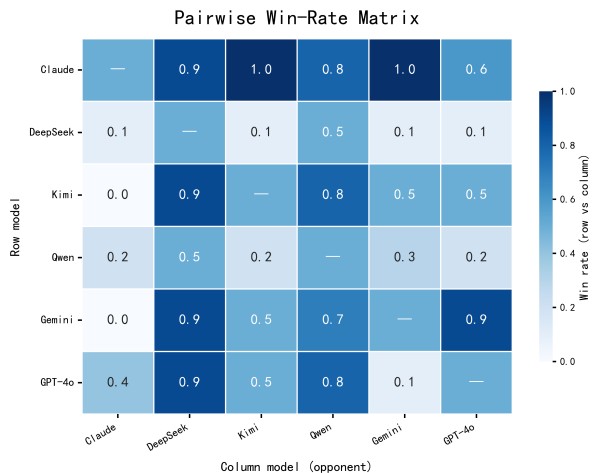

*Figure 2.* Pairwise head-to-head win-rate matrix among evaluated models. Each cell reports the empirical win rate of the row model against the column model (diagonal entries are not applicable).

A pilot test replacing precise coordinates with relative directional cues (Encirclement (Ambiguous), Table 3) reveals divergent sensitivity to missing spatial anchors: most top-tier models remain robust, while DeepSeek collapses with drastically increased variance (Appendix E).

**Front–Back Assault.** This heterogeneous-unit task yields a markedly different ranking. GPT-4o leads, followed closely by Kimi, which reverses its weak encirclement standing. Gemini and DeepSeek occupy a competitive mid-tier, while Claude—despite its encirclement strength—drops to fifth place. Qwen trails the field. The reversal of Claude and Kimi between encirclement and front–back assault underscores that attribute-aware role assignment and symmetric spatial geometry draw on partially disjoint capabilities.

**Visibility Coverage.** Claude and Gemini lead reconnaissance, both exceeding 0.80 with low variance, suggesting systematic exploration strategies. DeepSeek and GPT-4o form a solid mid-tier, while Kimi and Qwen lag, the latter showing the highest variance. Notably, Gemini achieves strong coverage despite its weak encirclement, reinforcing that open-area search and tight formation closure are distinct skills.

These results reveal pronounced capability fragmentation rather than uniform tactical intelligence. Claude dominates symmetric spatial tasks but falters on heterogeneous role assignment. Kimi exhibits the opposite profile, strong at attribute-aware positioning but weaker at homogeneous coordination. GPT-4o remains the most balanced across all three dimensions, never ranking below fourth. Gemini shows the strongest task-specific specialization—top-tier reconnaissance alongside bottom-tier encirclement—confirming that its spatial reasoning is optimized for coverage rather than closure. DeepSeek and Qwen occupy middle-to-lower tiers across tasks, with DeepSeek exhibiting more consistent cross-task transfer and Qwen showing the weakest overall profile with high variance in exploration.

### 4.4. Analysis of Three Tracks

The three tracks together provide a unified view of long-horizon agent behavior: performance is multi-dimensional and context-dependent, so a single global ranking is neither stable nor diagnostic. Track A stresses end-to-end win conversion under path dependence by evaluating against a fixed panel of stylized rule-based opponents, revealing whether an agent can maintain a coherent macro-to-tactical loop and translate economy, information, and combat advantages into decisive outcomes. Track B then probes opponent dependence under cross-play, showing that head-to-head strength can diverge from performance against scripted baselines once the opponent can adapt and counter-plan, and thus exposing matchup-sensitive competitiveness that is obscured by any fixed opponent set. Track C complements both by directly testing instruction grounding under three human tactical directives and scoring the realized behaviors with process-level metrics, making it possible to attribute failures to specific coordination and reconnaissance mechanisms rather than to the final outcome alone.

*Table 3.* Tactical benchmark performance across LLMs. The Encirclement (Ambiguous) column reports encirclement scores from a pilot test with underspecified directional cues.

| Model | Encirclement | Encirclement (Ambiguous) | Front–Back Assault | Visibility Coverage |
|---|---|---|---|---|
| Claude | **0.647 ± 0.068** | **0.746 ± 0.073** | 0.456 ± 0.088 | **0.833 ± 0.084** |
| DeepSeek | 0.600 ± 0.096 | 0.221 ± 0.307 | 0.465 ± 0.112 | 0.740 ± 0.141 |
| Kimi | 0.495 ± 0.141 | 0.380 ± 0.119 | 0.492 ± 0.092 | 0.692 ± 0.111 |
| Qwen | 0.579 ± 0.091 | 0.626 ± 0.046 | 0.416 ± 0.059 | 0.670 ± 0.152 |
| Gemini | 0.386 ± 0.085 | 0.357 ± 0.251 | 0.485 ± 0.111 | 0.801 ± 0.076 |
| GPT-4o | 0.624 ± 0.078 | 0.624 ± 0.165 | **0.504 ± 0.095** | 0.738 ± 0.094 |

An important implication is that the identity of the strongest agent is task-dependent. Claude excels in symmetric spatial reasoning and competitive play against both rule-based and adaptive opponents, yet exhibits limitations in heterogeneous-unit role assignment. Kimi demonstrates the opposite pattern: its attribute-aware coordination in front–back assault and competitive head-to-head performance contrast with weaker encirclement and rule-based win conversion, suggesting that heterogeneous coordination emerges more effectively against adaptive opponents than under static scripts. GPT-4o maintains the most balanced profile across all three tracks, with competitive win conversion, resilient cross-play, and leading front–back assault performance, indicating broad robustness without clear dominance in any single regime. Gemini shows strong specialization toward information gathering and reconnaissance, alongside top-tier cross-play competitiveness, but struggles with tight formation closure and rule-based win conversion. DeepSeek and Qwen occupy lower tiers across tracks, with DeepSeek exhibiting more consistent execution and Qwen showing higher variance in tactical performance.

All main experiments use a standard prompt with full state descriptions and strategic guidance; a prompt-sensitivity ablation confirms that simplified prompts degrade performance (Appendix C). These results motivate multi-axis, instrumented evaluation as a prerequisite for developing LLM agents with reliable long-horizon competence.

These results motivate multi-axis, instrumented evaluation as a prerequisite for developing LLM agents with reliable long-horizon competence.

## 5. Conclusion

We introduce a reproducible benchmark for long-horizon adversarial decision-making that evaluates LLM agents as closed-loop decision modules under partial observability, strategic interaction, and delayed consequences. Our three-track protocol shows that model rankings vary substantially across opponents, interaction regimes, and human instructions, indicating that single aggregate scores can conceal interaction-specific weaknesses. These findings highlight the value of diagnostic, process-oriented evaluation and point to two directions for future progress: opponent-aware adaptive decision-making and human-in-the-loop hierarchical control. While our benchmark provides a controlled diagnostic setting, real-world steerability, potential pretraining contamination, and the broader validity of process-level metrics remain open directions for future validation.

## Impact Statement

This work is expected to have a primarily scientific and governance-oriented impact by improving the transparency and reproducibility of evaluating LLM agents in long-horizon interactive settings. By making long-horizon failure modes more observable and comparable, the benchmark can support more responsible deployment decisions and reduce reliance on outcome-only claims that may mask brittle behavior. Potential negative impacts include dual-use: evaluation protocols and diagnostics may be repurposed to systematically optimize adversarial agents, and benchmark-driven development may incentivize overfitting to a narrow testbed rather than improving general reliability. There are also broader community implications around access and compute, since results can reflect differences in model availability and evaluation resources. We partially mitigate these concerns by emphasizing measurement and attribution, highlighting brittleness and exploitability through process-level diagnostics, and encouraging transparent reporting and cautious interpretation of benchmark results.

## Limitations

**Human steerability evaluation.** Our current Track C design prioritizes experimental control and reproducibility by using standardized, pre-triggered instructions. This does not fully capture the ambiguity, dynamic revision, or erroneous instructions common in real human-AI interactions (e.g., humans adjusting tactics midway or describing strategies with ambiguous language). Nor does it evaluate LLMs' ability to identify, reject, or correct unreasonable instructions. Consequently, Track C results should be interpreted as a controlled diagnostic of command-grounding capability rather

than a comprehensive measure of real-world steerability. We provide an extended evaluation protocol in Appendix E for future work, specifying test scenarios with ambiguous instructions, dynamically revised instructions, and deliberately erroneous instructions to evaluate model robustness.

**Pretraining overlap.** We cannot fully rule out RTS-related public data (e.g., strategy guides, replays, community discussions) in pretraining corpora. However, all models are evaluated under the same interface, rules, map, and budget, and we observe strong cross-track ranking reversals and model-specific strengths that are difficult to explain by memorization alone. Even if general RTS prior exists in model training, the observed cross-model, cross-track performance disparities remain valid and informative for diagnosing LLM long-horizon decision-making competencies.

**Metric design trade-offs.** We do not perform early-/mid-/late-game temporal decomposition because phase boundaries are not well-defined in our free-form annihilation setup; both sides start with 20k credits and can freely allocate resources, making fixed temporal splits arbitrary. Capability-based failure localization (scouting, combat efficiency, resource use, command following) is more appropriate here. Additionally, while we report sensitivity analyses for C1–C3 metrics (Kendall's $\tau = 0.819, 0.800, 0.828$ across parameter settings; Appendix F), these measures remain proxies that should be validated against broader outcome sets in future work.

**Scope.** This benchmark focuses on within-game long-horizon planning. Cross-game adaptation, persistent memory, and learning across matches are explicitly out of scope and positioned as future work.

## Acknowledgements

We thank members of the State Key Laboratory of Complex Product Intelligent Manufacturing System Technology for helpful discussions.This work was supported in part by the National Natural Science Foundation of China under Grant 62192751, in part by the Key R&D Project of China under Grant 2017YFC0704100, in part by the 111 International Collaboration Program of China under Grant B25027, in part by BNRist Program under Grant BNR2019TD01009, in part by the InnoHK Initiative, The Government of HKSAR, and in part by the Laboratory for AI-Powered Financial Technologies.

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

# A. Tactical Benchmark Visualizations

This appendix presents visual examples of the three tactical benchmark scenarios evaluated in our experiments. Each figure shows representative execution traces from different LLM agents.

## A.1. Encirclement (Pincer Attack) Scenarios

Figures 3–8 illustrate the temporal evolution of encirclement formations executed by the six LLMs. Each figure displays three sequential snapshots (left to right) showing the progression from initial positioning through intermediate maneuvering to final surrounding formation. Red units represent the attacking force coordinated by the LLM, while blue units represent the target formation.

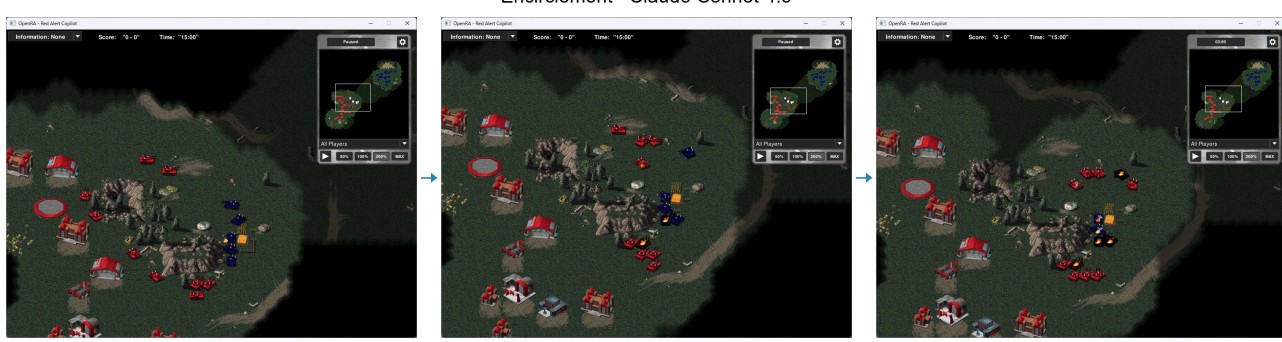

*Figure 3.* Encirclement execution by Claude (Enc $= 0.647 \pm 0.068$). The agent demonstrates the strongest spatial coordination among all models, achieving optimal attack distance while maintaining even angular distribution around the target with low run-to-run variance.

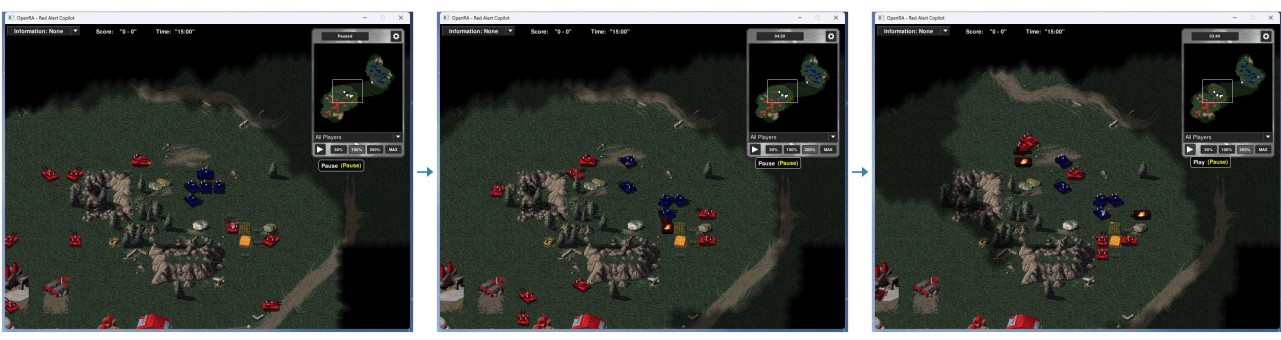

*Figure 4.* Encirclement execution by GPT-4o (Enc $= 0.624 \pm 0.078$). The agent demonstrates strong spatial coordination with balanced proximity constraints and angular diversity across trials.

## A.2. Front–Back Formation (Role-Constrained Coordination) Scenarios

Figures 9–14 demonstrate role-based unit coordination with heterogeneous forces. Each LLM controls heavily-armored super-heavy tanks (high HP, low damage, larger red units) and fragile V2 rocket launchers (low HP, high damage, smaller red units). Optimal coordination requires positioning tanks at the front to absorb damage while preserving launchers in the rear. The three frames per figure show the temporal evolution of formation adjustment.

## A.3. Exploration Rate (Reconnaissance) Comparison

Figure 15 presents a direct comparison of reconnaissance performance across all six LLMs. Each panel shows the final exploration coverage achieved by V2 rocket launcher units (red) after a fixed time period. Darker areas indicate unexplored regions, while lighter areas show revealed terrain. The exploration rate (ER) quantifies the percentage of the map successfully scouted.

Encirclement - DeepSeek-V3.2

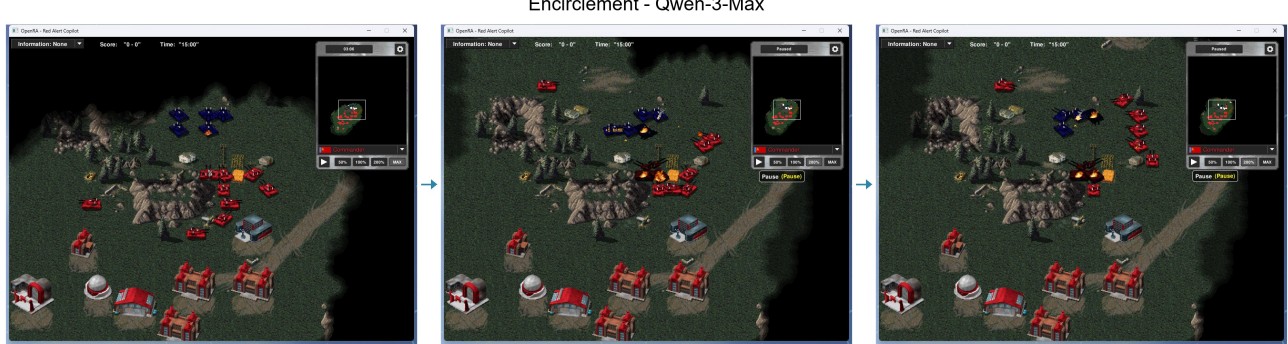

*Figure 5.* Encirclement execution by DeepSeek (Enc $= 0.600 \pm 0.096$). Similar to GPT-4o, DeepSeek achieves strong encirclement quality through coordinated multi-directional approach, though with slightly higher variance.

Encirclement - Qwen-3-Max

*Figure 6.* Encirclement execution by Qwen (Enc $= 0.579 \pm 0.091$). Qwen demonstrates moderate encirclement capability with noticeable gaps in angular coverage compared to the top-tier models.

### A.4. Model Comprehensive Ability (Track A Summary)

Figure 16 provides a radar-chart summary of the six evaluated LLMs across five Track A dimensions: win robustness (mean win rate), combat efficiency (KLR), economic throughput (MC), resource utilization (MU), and exploration capability (Scouting). The visualization highlights the sharp specialization and cross-style trade-offs discussed in Sec. 3.2, with Claude and GPT-4o occupying complementary outer envelopes while Kimi and DeepSeek cluster toward the center.

### A.5. Key Observations from Visual Analysis

Task-Specific Visual Patterns. The figures reveal distinct capability profiles across models and scenarios:

- **Homogeneous coordination (Encirclement):** Claude, GPT-4o, and DeepSeek produce visually similar surrounding formations with even angular distribution, while Kimi shows moderate but inconsistent positioning, and Gemini exhibits clustered positioning patterns with incomplete coverage.

- **Heterogeneous coordination (Front–Back):** GPT-4o and Kimi demonstrate clear spatial separation between unit types, whereas DeepSeek and Claude exhibit mixed positioning despite strong encirclement performance. Qwen shows the weakest role differentiation overall.

- **Exploration:** Claude and Gemini show systematic dispersed coverage patterns, while DeepSeek and GPT-4o exhibit moderate search with some revisited areas. Kimi and Qwen show more concentrated or incomplete coverage, with Qwen leaving significant unexplored pockets.

Variance Manifestation. High-variance models show qualitatively different execution patterns across trials rather than minor variations in the same strategy. For example, Gemini's front-back formations range from near-optimal role separation

Encirclement - Kimi-K2.5-FW

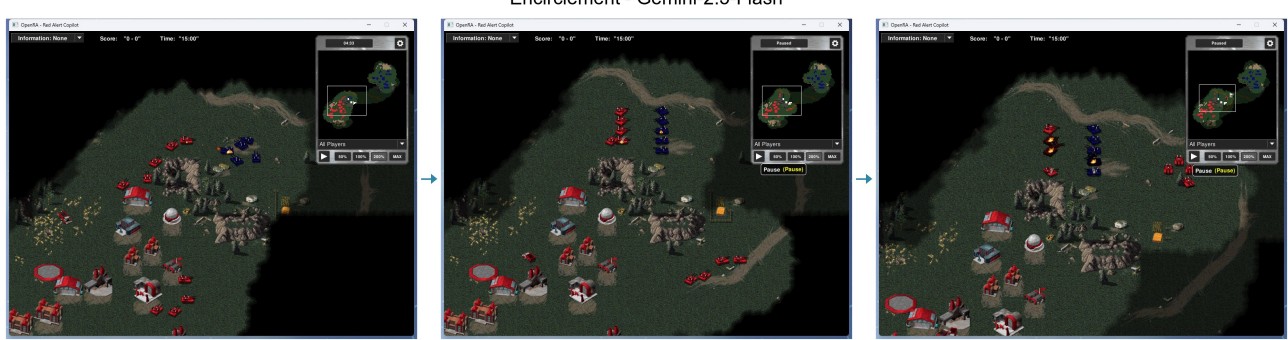

*Figure 7.* Encirclement execution by Kimi (Enc $= 0.495 \pm 0.141$). The agent shows inconsistent spatial distribution with high run-to-run variance, occasionally leaving exploitable gaps in the surrounding geometry.

Encirclement - Gemini-2.5-Flash

*Figure 8.* Encirclement execution by Gemini (Enc $= 0.386 \pm 0.085$). Gemini exhibits significant challenges in forming effective surrounding formations, with poor angular distribution and units concentrated in limited sectors.

to complete role inversion, explaining the high standard deviation (std=0.111). Kimi similarly shows high variance in encirclement (std=0.141), with some trials achieving adequate coverage and others leaving critical angular gaps.

**Variance Manifestation.** High-variance models show qualitatively different execution patterns across trials rather than minor variations in the same strategy. For example, Gemini's front-back formations range from near-optimal role separation to complete role inversion, explaining the high standard deviation (std=0.220).

**Capability Gaps vs. Execution Errors.** DeepSeek's consistent failure in role-constrained coordination (low FB score with low variance) manifests visually as systematically incorrect positioning rather than random errors. This contrasts with Qwen's high-variance performance, which shows correct patterns in some frames but poor execution in others, suggesting inconsistent strategy rather than fundamental incapability.

## B. Metric Correlation and Coupling Analysis

We conducted correlation and multivariate analyses between process-level metrics and win rate. Table 4 reports univariate Pearson correlations, and Table 5 reports standardized coefficients from a four-metric multivariate regression.

The main finding is that no single metric is universally decisive. In Track A, MU has the highest univariate correlation ($r = 0.95$), but its independent contribution becomes very small in multivariate regression ($\beta = 0.012$), because it is strongly coupled with KLR ($r = 0.73$) and Scouting ($r = 0.75$). Thus, MU is better viewed as a proxy for a broader capability cluster combining combat efficiency and information advantage, rather than as an independent driver. The joint effect of metrics is much more informative than any single one: the four-metric model achieves $R^2 = 0.972$ in Track A and 0.994 in Track B. KLR and Scouting are consistently helpful across tracks, while MU is more predictive in Track A and MC is relatively weak. In Track B, Scouting becomes the strongest factor ($\beta = 0.238$) and MU turns negative ($\beta = -0.152$),

Front–Back Formation - GPT-4o

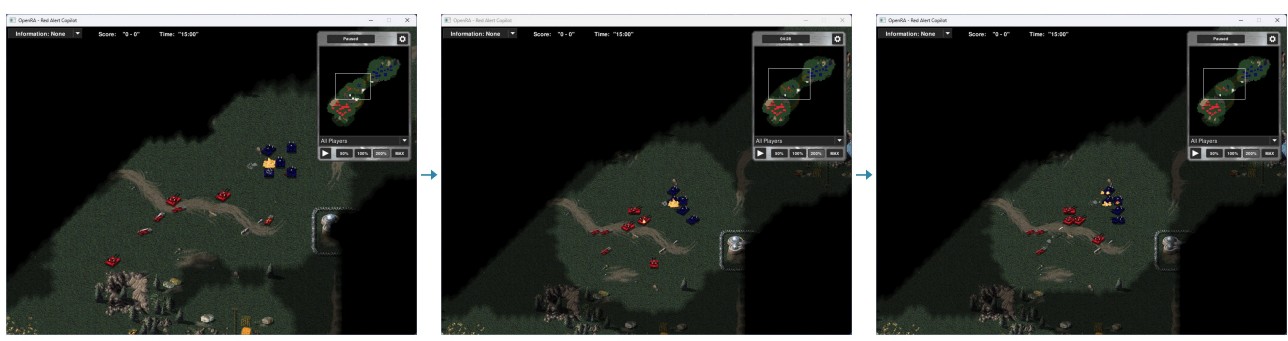

*Figure 9.* Front-back coordination by GPT-4o (FB $= 0.504 \pm 0.095$). GPT-4o demonstrates superior attribute-aware positioning, consistently placing super-heavy tanks closer to enemy positions while maintaining protected rear positions for V2 launchers.

Front–Back Formation - Kimi-K2.5-FW

*Figure 10.* Front-back coordination by Kimi (FB $= 0.492 \pm 0.092$). Kimi demonstrates surprisingly robust role differentiation given its weaker encirclement performance, with effective spatial separation between unit types.

suggesting that against adaptive opponents, information advantage matters more than fine-grained resource optimization. Overall, victory arises from the interaction of combat, economy, and scouting rather than from optimizing one metric alone.

## C. Prompt Sensitivity Ablation

To assess sensitivity to prompt design, we compared two prompt variants: a standard version with full state descriptions and strategic guidance, and a simplified version with only the JSON constraint and a minimal task description.

The simplified prompt leads to a clear drop in win rate. Without explicit subgoal decomposition and reasoning anchors, the model becomes less stable in balancing economic management and military actions. This suggests that, in complex RTS tasks, current LLMs still benefit from structured domain guidance to maintain consistent long-horizon decision-making. All main experiments use the empirically more stable standard prompt, so the reported results primarily reflect the method itself rather than prompt engineering effects.

## D. Statistical Stability Analysis

We quantify the sensitivity of reported rankings to sample size through two complementary analyses: a 15-run repeat experiment for Track A, and a pair-stratified bootstrap for Track B.

**15-run repeat (Track A).** We repeated the Rush AI evaluation 15 times and analyzed win rate, combat efficiency, macro economy, and map exploration. The 15-run results recover the same overall ordering (GPT-4o > Gemini > Qwen > DeepSeek) and the same relative strengths, suggesting that five runs are sufficient to recover stable comparative trends in our setup.

Front–Back Formation - Gemini-2.5-Flash

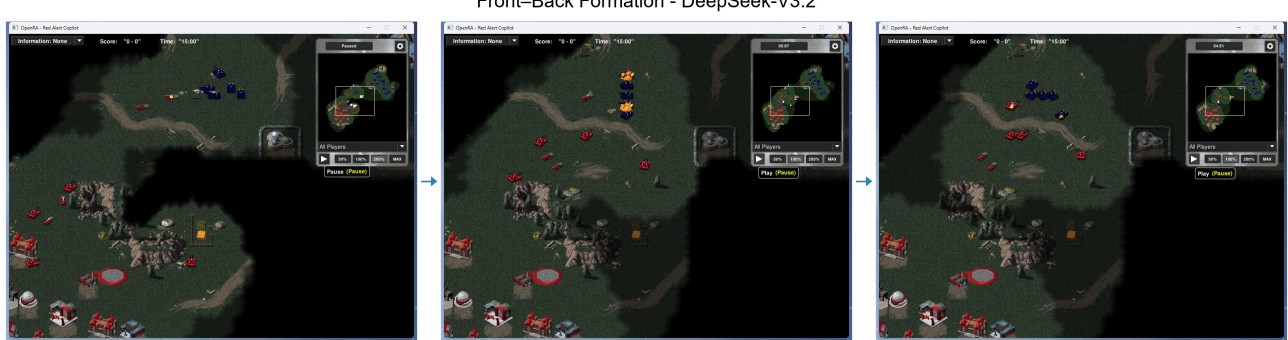

*Figure 11.* Front-back coordination by Gemini (FB $= 0.485 \pm 0.111$). Gemini achieves competitive average performance but with the highest variance among all models, alternating between coherent formations and abrupt breakdowns.

Front–Back Formation - DeepSeek-V3.2

*Figure 12.* Front-back coordination by DeepSeek (FB $= 0.465 \pm 0.112$). Despite strong encirclement performance, DeepSeek exhibits weak role-constrained coordination, with poor differentiation between tank and launcher positioning.

**Pair-stratified bootstrap (Track B).** We resampled 5 games with replacement per pair over 1000 replicates and refit the Davidson model. Table 7 reports bootstrapped model scores and percentile 95% confidence intervals. The top of the ranking is very stable: the 95% CI for Gemini is $[1927.51, 2766.36]$, and Gemini outranks GPT-4o in 99.1% of replicates; the 95% CI for GPT-4o is $[1508.05, 1961.52]$, entirely above zero. The less certain separation is between Qwen and DeepSeek: the 95% CI for their score difference is $[-75.65, 726.14]$, crossing zero, with Qwen outranking DeepSeek in 93.3% of replicates.

Table 8 reports bootstrapped pairwise score differences. The top-two ordering is robust; the Qwen–DeepSeek separation is comparatively more uncertain.

**Subsampling stability.** Table 9 reports pair-stratified $n$-of-5 subsampling results. The ranking stabilizes rapidly: with $n = 2$, the exact full-data ranking is recovered in 80.8% of subsamples; with $n = 3$, exact recovery reaches 93.5%; with $n = 4$, all 1000 subsamples recover the exact full-data ranking.

## E. Track C Extended Evaluation Protocol

Our current Track C uses standardized, pre-triggered instructions to ensure experimental control and reproducibility. To move toward more realistic human-AI interaction, we propose the following extended protocol for future work. The protocol introduces three dimensions of complexity absent from the current benchmark: ambiguous instructions, dynamically revised instructions, and erroneous instructions.

**E.1 Ambiguous instructions (pilot experiment).** We conducted a preliminary study on the Encirclement task with mild ambiguity: replacing precise coordinates with a relative directional cue (e.g., "in the open ground northeast of our base"). As reported in Table 3 (Encirclement (Ambiguous) column), most models exhibit divergent sensitivity to missing spatial

Front–Back Formation - Claude-Sonnet-4.5

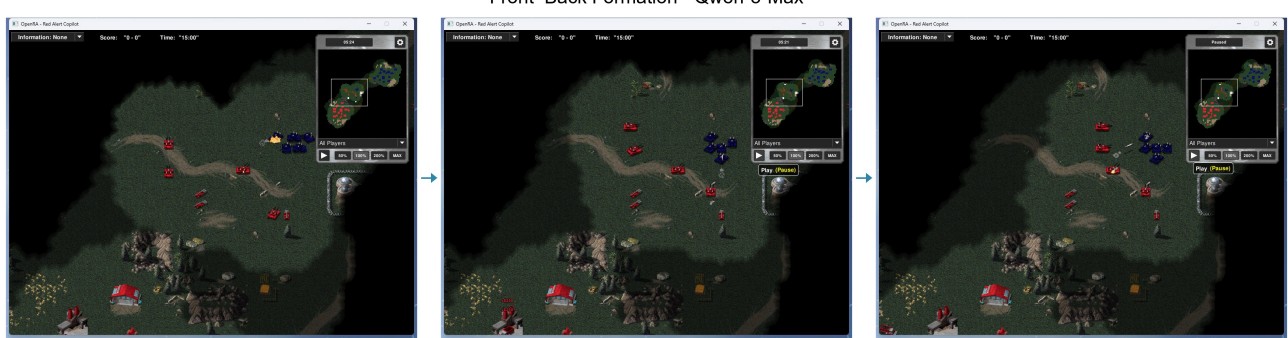

*Figure 13.* Front-back coordination by Claude (FB $= 0.456 \pm 0.088$). Despite leading in encirclement and reconnaissance, Claude degrades noticeably in heterogeneous assault, indicating weaker attribute-aware reasoning.

Front–Back Formation - Qwen-3-Max

*Figure 14.* Front-back coordination by Qwen (FB $= 0.416 \pm 0.059$). Qwen trails the field with the weakest role differentiation, though with relatively low variance indicating systematic rather than stochastic limitation.

anchors. Claude, Qwen, and GPT-4o remain largely unaffected, suggesting robust geometric reasoning under underspecified conditions. Kimi and Gemini degrade noticeably, reflecting heavier reliance on precise spatial anchors. DeepSeek suffers the most severe collapse alongside drastically increased variance; log review reveals frequent failure to infer enemy location from vague directional cues, resulting in multiple zero-score trials where units wander without engaging. These results suggest that strong explicit performance on homogeneous coordination does not guarantee robustness to instruction ambiguity.

**E.2 Dynamically revised instructions.** The operator issues an initial command, then mid-execution revises the objective (e.g., switching from encirclement to frontal assault at 50% completion). The agent must detect the revision, abandon the prior plan, and re-coordinate units under new constraints. Evaluation measures transition latency, formation disruption, and final compliance with the revised intent.

**E.3 Erroneous instructions.** The operator deliberately issues a tactically unsound command (e.g., ordering all units to attack a heavily fortified position without reconnaissance). The agent must identify the error, reject or question the command, and propose an alternative. Evaluation measures error detection rate, justification quality, and safety of the executed alternative.

These extensions are provided as protocol specifications; full implementation and integration into the benchmark are left to future work.

# F. Metric Robustness Validation (C1–C3)

To verify that the Track C tactical metrics (C1 Encirclement, C2 Front–Back Assault, C3 Visibility Coverage) are robust to parameter choices and implementation details, we performed sensitivity analyses using Kendall's $\tau$ over reasonable parameter settings.

**Visibility Coverage Comparison**

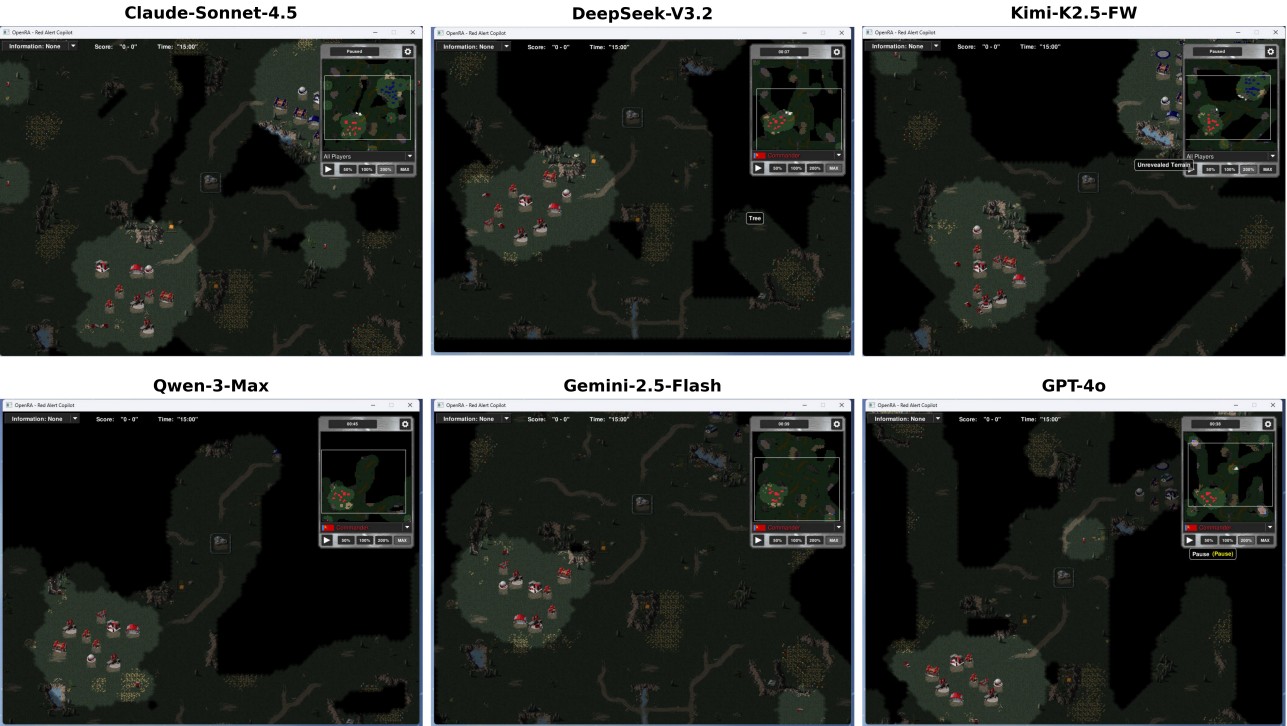

*Figure 15.* Exploration rate comparison across six LLMs. Top-left: Claude (VC=0.833); Top-middle: Gemini (VC=0.801); Top-right: DeepSeek (VC=0.740); Bottom-left: GPT-4o (VC=0.738); Bottom-middle: Kimi (VC=0.692); Bottom-right: Qwen (VC=0.670).

*Table 4.* Correlation between process-level metrics and win rate.

| METRIC | DESCRIPTION | TRACK A | TRACK B |
|---|---|---|---|
| KLR | KILL-LOSS RATIO | $r = 0.78^*$ | $r = 0.79^*$ |
| MC | MINERAL INCOME PER MINUTE | $r = 0.54$ | $r = -0.33$ |
| MU | MILITARY UTILITY (TOTAL EXPENDITURE) | $r = 0.95$ | $r = 0.67$ |
| SCOUTING | EXPLORATION SCORE | $r = 0.79^*$ | $r = 0.86^*$ |

**C1 Encirclement.** We varied the engagement radius $d_{\min}$, the participation decay rate $\alpha$, and the coverage sensitivity $\beta$ across $\pm 20\%$ of their default values. The Kendall correlation between rankings under perturbed settings and the default setting is $\tau = 0.819$.

**C2 Front–Back Assault.** We varied $d_{\min}$, $\alpha$, and the discordance tolerance $\varepsilon$ across $\pm 20\%$. The resulting rank correlation is $\tau = 0.800$.

**C3 Visibility Coverage.** We varied the horizon length $\Delta$ and grid discretization granularity across $\pm 20\%$. The resulting rank correlation is $\tau = 0.828$.

These values indicate that model rankings remain stable across reasonable configuration changes rather than depending on one specific parameter choice.

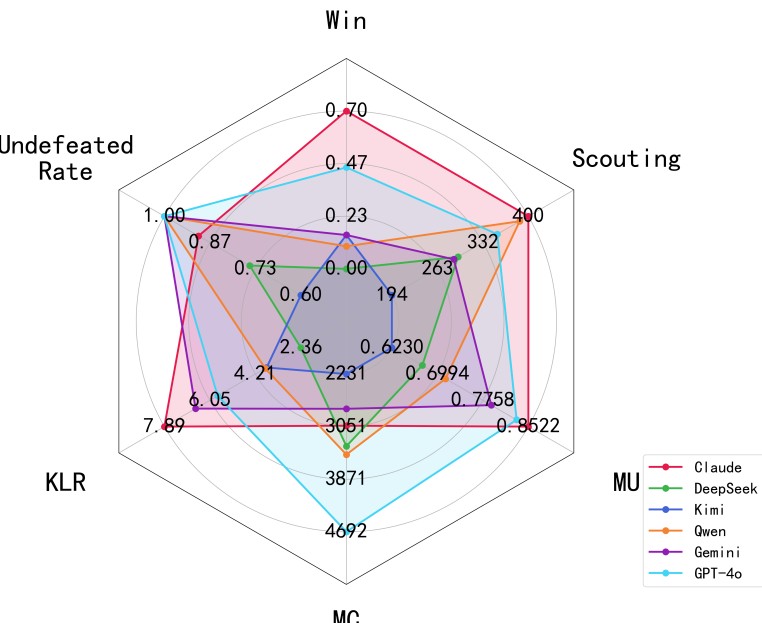

*Figure 16.* Model comprehensive ability comparison across six LLMs on Track A. Radar chart summarizing win robustness, combat efficiency, economic throughput, resource utilization, and exploration capability.

*Table 5.* Multivariate regression of win rate on four process-level metrics.

| TRACK | $R^2$ | ADJ. $R^2$ | KEY DRIVERS ($\beta$) |
|---|---|---|---|
| TRACK A | 0.972 | 0.944 | KLR (0.097) > MC (0.05) > SCOUTING (0.046) > MU (0.012) |
| TRACK B | 0.994 | 0.988 | SCOUTING (0.238) > MU (−0.152) > KLR (0.128) > MC (0.043) |

## G. System Prompt for Tracks A and B

The following is the complete system prompt used for all LLM agents in Tracks A and B. The two tracks share an identical prompt; the only difference is the opponent type (rule-based AI in Track A, peer LLM agents in Track B). Track C employs a structurally identical prompt with modified objective clauses tailored to human tactical directives; this variant is provided in the supplementary repository.

```
SYSTEM_PROMPT = """
You are the sole decision-making core of the OpenRA agent.
Your only behavior is to generate a one-step action list (JSON) based on the observed state.

[State Description]
Each observation includes:
- cycle (round number), money (funds), power (usage/supply)
- my_buildings (key structures: refineries, war factories, tech buildings, etc.)
- combat_units (friendly combat units: tanks, anti-air vehicles, construction yards, etc.,
  including id, type, hp, pos)
- harvesters (mining vehicle statistics: total count, average hp, damaged list with id/hp/pos)
- power_buildings (power plant statistics: total count, total output, damaged list with id/hp/pos)
- enemies (visible enemy units, including id, type, hp, pos)
- can_produce (currently available production list)

Note: To optimize context length, harvesters and power plants are aggregated;
only damaged units list detailed information (id/hp/pos).

[Objectives]
1 Maximize fund acquisition efficiency while quickly spending funds to build combat units
   to strike enemy forces and territories;
2 Explore the entire map to gradually expand visible territory;
```

*Table 6.* Prompt sensitivity ablation (GPT-4o vs. rule-based opponents). W/D/L denotes win, draw, and loss rate.

| OPPONENT | STANDARD W/D/L | SIMPLIFIED W/D/L |
|---|---|---|
| OVERALL | 0.15 / 0.85 / 0 | 0 / 0.59 / 0.41 |
| NAVAL AI | 0 / 1 / 0 | 0 / 1 / 0 |
| NORMAL AI | 0.4 / 0.6 / 0 | 0 / 0.67 / 0.33 |
| RUSH AI | 0.2 / 0.8 / 0 | 0 / 0 / 1 |
| TURTLE AI | 0 / 1 / 0 | 0 / 0.67 / 0.33 |

*Table 7.* Bootstrapped model scores and 95% CIs (1000 replicates).

| MODEL | SCORE | 95% CI | TOP RANK (PROB.) |
|---|---|---|---|
| GEMINI | 2180.66 | [1927.51, 2766.36] | 1 (99.1%) |
| GPT-4O | 1689.58 | [1508.05, 1961.52] | 2 (99.1%) |
| QWEN | 1173.34 | [857.98, 1354.67] | 3 (93.3%) |
| DEEPSEEK | 956.42 | [409.45, 1159.30] | 4 (93.3%) |

```
3 Avoid power deficits (power usage must not exceed supply) and maintain production rhythm;
4 If enemies are detected, concentrate firepower to strike;
5 Annihilate all enemy targets.

Hard Rules (comply strictly):
- Output valid JSON only. Required keys: build, unit_commands, notes.
- Never repeat or plagiarize example content; examples illustrate format only, not strategy.
- Decide solely based on the observed state. If a unit/structure is unavailable,
  choose not to produce it or build prerequisite technology first.
- Maximum 40 unit_commands per cycle to avoid overload and JSON truncation.
- unit_commands may only operate on combat units listed in state.combat_units (by actor_id).
- Attack and lock-on targets are limited to visible enemies in state.enemies (by id).
- If no action is necessary, return empty lists (e.g., []).

Permitted Actions (Action Space):
- Build/Produce: name_or_code (Chinese building name or unit code), count,
  is_building (True = building, False = unit)
- Unit Commands:
  - move: move unit to target coordinates target{x,y}
  - attack: assign a single visible enemy target_id for the unit to attack
  - attack_move: move unit to target coordinates target{x,y}; unit may attack
    enemies encountered en route

Strategic Hints (optional):
- If harvesting is weak: prioritize ore refineries and harvesters. One refinery can
  support multiple harvesters, but ideally fewer than three per refinery.
- If power is insufficient: prioritize power plants.
- To unlock vehicle production: build war factories / workshops.
- Advanced vehicle production requires higher-tier structures: radar, service depots,
  nuclear plants, and tech centers.
- Exploration: dispatch scouts (jeeps / apc / ftrk) to different sectors or grid points
  to gradually scout the map.
- Combat: when encountering threats, support units may use attack_move toward
  the threat vicinity.

Output Format (schema, for field definition only; do not copy example values):

#####Action#####
{
  "build": [
    {"name_or_code":"<string>", "count":<<int>, "is_building":<<bool>, "why":"<string-optional>"}
  ],
  "unit_commands": [
```

*Table 8.* Bootstrapped pairwise score differences.

| PAIR | POINT EST. | 95% CI | PROB. $> 0$ |
|---|---|---|---|
| GEMINI VS GPT-4O | 491.07 | [126.13, 1131.19] | 99.1% |
| GPT-4O VS QWEN | 516.24 | [210.06, 1014.80] | 100.0% |
| QWEN VS DEEPSEEK | 216.92 | [−75.65, 726.14] | 93.3% |
| GEMINI VS QWEN | 1007.31 | [677.49, 1739.22] | 100.0% |
| GEMINI VS DEEPSEEK | 1224.24 | [826.08, 2345.43] | 100.0% |
| GPT-4O VS DEEPSEEK | 733.17 | [434.29, 1424.20] | 100.0% |

*Table 9.* Pair-stratified $n$-of-5 subsampling (1000 replicates per $n$).

| $n$ | EXACT RANKING | TOP-TWO SET | MEAN ABS. DEV. |
|---|---|---|---|
| 2 | 80.8% | 97.4% | 893.7 |
| 3 | 93.5% | 100.0% | 114.0 |
| 4 | 100.0% | 100.0% | 64.1 |
| 5 | 100.0% | 100.0% | 0.0 |

```
    {"actor_id":<<int>, "action":"move|attack|attack_move",
      "target":{"x":<<int>,"y":<<int>}?, "target_id":[<<int>]?}
  ],
  "notes":"<string-optional>"
}

Do not output any text outside JSON.
"""
```

