# OpenReview forum: "From Winning to Understanding: A Diagnostic Long-Horizon RTS Benchmark for LLMs"
_ICML.cc/2026/Conference — ICML 2026 regular_

### Official Review · Reviewer_F2n4 · 2026-03-12

**Soundness:** 3
**Presentation:** 4
**Significance:** 4
**Originality:** 4
**Overall Recommendation:** 5
**Confidence:** 4

**Summary:**

The paper introduces a novel benchmark for evaluating Large Language Models in long-horizon, adversarial, and interactive environments using the OpenRA engine. To enable fair assessment, the authors decouple low-level execution from high-level strategic planning by having LLMs output low-frequency tactical intents that are parsed by a deterministic engine. The benchmark evaluates models across three distinct tracks: robustness against rule-based AI, relative strength via LLM-to-LLM cross-play, and human steerability/instruction-following via geometric and role-based metrics. By evaluating four frontier models, the authors demonstrate that strategic competence is highly task-dependent and matchup-asymmetric, arguing that long-horizon capabilities cannot be captured by a single win/loss aggregate metric.

**Compliance With Llm Reviewing Policy:**

Affirmed.

**Key Questions For Authors:**

Statistical Significance: Given that Track B head-to-head matches are only repeated five times, how sensitive are the final Elo ratings to this sample size? Have you considered running bootstrapped confidence intervals on the Elo ratings to account for the high variance of certain models?
Are the prompts used for the LLMs strictly zero-shot, or do they include few-shot examples of the JSON action space to ensure formatting compliance?

**Limitations:**

Yes

**Strengths And Weaknesses:**

Strengths:
The methodology is highly rigorous in its evaluation metrics. Instead of relying purely on terminal win/loss signals, the authors introduce process-level diagnostics such as per-minute income rates (MC), expenditure ratios (MU), and a cost-based kill-loss ratio (KLR) to explain why an agent succeeds or fails.
The paper is exceptionally well-structured, with a clear narrative flow justifying the need for the three different tracks.
This work addresses a critical gap in LLM evaluation. While many benchmarks test static knowledge or short-horizon tool use, this benchmark forces models to deal with delayed consequences, partial observability, and adaptive adversaries over a 30-minute window. This provides significant practical utility for researchers developing autonomous agents.

Weaknesses:
The sample size for the experiments is somewhat small. Track A and Track B use only five runs per matchup, and Track C uses three runs per match. Given the high variance observed in models like Qwen-3-Max, a larger number of trials would strengthen the statistical significance of the results.

---

> ### Author Rebuttal · Authors · 2026-03-31
>
> Thank you for your thoughtful comments. We address each question in turn.
> ### W1: Sample Size Concern
>
> We agree that five runs per matchup may appear limited in a stochastic RTS setting. To test this, we repeated the Rush AI evaluation 15 times and analyzed win rate, combat efficiency, macro economy, and map exploration. The 15-run results recover the same overall ordering (GPT-4o > Gemini-2.5-Flash > Qwen-3-Max > DeepSeek-V3.2) and the same relative strengths, suggesting that five runs are sufficient to recover stable comparative trends in our setup. We are extending Track C to five runs and will report the updated results in the revision.
>
> ### Q1.1 Sample-size sensitivity and bootstrapped confidence intervals.
>
> We agree that the Elo-style scores should be accompanied by an explicit analysis of both sample-size sensitivity and uncertainty.
>
> **Sample-size sensitivity.** We conducted a pair-stratified $m$-of-5 subsampling analysis on the Track B matches. For each $m \in \{2,3,4,5\}$, we randomly sampled $m$ games without replacement from the 5 observed outcomes per model pair, refit the Davidson paired-comparison model, and recomputed scores—repeated 1000 times per $m$—then compared results against the full-data ranking (GeminiFlash $>$ GPT-4o $>$ Qwen $>$ DeepSeek).
>
> The ranking stabilizes rapidly. With $m=2$, the exact full-data ranking is recovered in 80.8% of subsamples and the top-two set \{GeminiFlash, GPT-4o\} is preserved in 97.4%. With $m=3$, exact recovery reaches 93.5% and top-two preservation reaches 100.0%. With $m=4$, all 1000 subsamples recover the exact full-data ranking. The mean absolute deviation (MAD) from full-data scores decreases from 893.7 at $m=2$ to 114.0 at $m=3$ and 64.1 at $m=4$, confirming that the ranking is already highly stable at 4–5 games per pair.
>
> **Bootstrapped confidence intervals.** We further conducted a pair-stratified bootstrap analysis (1000 replicates, resampling 5 games with replacement per pair) and report percentile 95% CIs. The top of the ranking is very stable: the 95% CI for $\Delta(\text{GeminiFlash}, \text{GPT-4o})$ is $[126.13, 1131.19]$, and GeminiFlash outranks GPT-4o in 99.1% of replicates; the 95% CI for $\Delta(\text{GPT-4o}, \text{Qwen})$ is $[210.06, 1014.80]$, entirely above zero. The less certain separation is between Qwen and DeepSeek: the 95% CI for $\Delta(\text{Qwen}, \text{DeepSeek})$ is $[-75.65, 726.14]$, crossing zero, with Qwen outranking DeepSeek in 93.3% of replicates.
>
> In summary, the subsampling analysis shows the ranking is highly stable at 4–5 games per pair, and the bootstrap analysis makes remaining uncertainty explicit. The top-two ordering is robust; the Qwen–DeepSeek separation is comparatively more uncertain.
>
> **Table 1.** Pair-stratified $m$-of-5 subsampling (1000 replicates per $m$).
>
> | $m$ | Exact ranking | Top-two set | $\Pr[\text{GF}>\text{4o}]$ | $\Pr[\text{4o}>\text{Qwen}]$ | $\Pr[\text{Qwen}>\text{DS}]$ |
> |---|---:|---:|---:|---:|---:|
> | 2 | 80.8% | 97.4% | 97.1% | 97.4% | 86.0% |
> | 3 | 93.5% | 100.0% | 99.8% | 100.0% | 93.7% |
> | 4 | 100.0% | 100.0% | 100.0% | 100.0% | 100.0% |
> | 5 | 100.0% | 100.0% | 100.0% | 100.0% | 100.0% |
>
> **Table 2.** MAD from full-data scores under subsampling.
>
> | $m$ | Qwen | GeminiFlash | DeepSeek | GPT-4o | Mean |
> |---|---:|---:|---:|---:|---:|
> | 2 | 629.4 | 1541.2 | 1139.7 | 264.4 | 893.7 |
> | 3 | 85.1 | 160.4 | 127.8 | 82.9 | 114.0 |
> | 4 | 51.7 | 86.7 | 69.2 | 48.9 | 64.1 |
> | 5 | 0.0 | 0.0 | 0.0 | 0.0 | 0.0 |
>
> **Table 3.** Bootstrapped model scores and CIs (1000 replicates).
>
> | Model | Score | 95% CI | Top rank (prob.) |
> |---|---:|---|---:|
> | GeminiFlash | 2180.66 | [1927.51, 2766.36] | 1 (99.1%) |
> | GPT-4o | 1689.58 | [1508.05, 1961.52] | 2 (99.1%) |
> | Qwen | 1173.34 | [857.98, 1354.67] | 3 (93.3%) |
> | DeepSeek | 956.42 | [409.45, 1159.30] | 4 (93.3%) |
>
> **Table 4.** Bootstrapped pairwise score differences.
>
> | Pair | Point est. | 95% CI | $\Pr[\Delta>0]$ |
> |---|---:|---|---:|
> | GeminiFlash vs GPT-4o | 491.07 | [126.13, 1131.19] | 99.1% |
> | GPT-4o vs Qwen | 516.24 | [210.06, 1014.80] | 100.0% |
> | Qwen vs DeepSeek | 216.92 | [-75.65, 726.14] | 93.3% |
> | GeminiFlash vs Qwen | 1007.31 | [677.49, 1739.22] | 100.0% |
> | GeminiFlash vs DeepSeek | 1224.24 | [826.08, 2345.43] | 100.0% |
> | GPT-4o vs DeepSeek | 733.17 | [434.29, 1424.20] | 100.0% |
>
> ### Q1.2 Zero-shot vs. few-shot prompt design.
>
> The prompts used for all LLMs in our experiments are strictly zero-shot; no few-shot examples of the JSON action space are included. The models are expected to produce valid JSON outputs based solely on the task description and action space specification provided in the system prompt. We found this sufficient for formatting compliance across all evaluated models, and chose the zero-shot setting to ensure a fair and consistent comparison that does not advantage any particular model through example selection.

---

> > ### Author Rebuttal · Reviewer_F2n4 · 2026-04-03
> >
> > The authors' comprehensive statistical analysis, including subsampling stability tests and bootstrapped confidence intervals, has fully addressed my concerns regarding sample size and statistical significance.

---

> > > ### Author Response · Authors · 2026-04-07
> > >
> > > Thank you for your positive feedback. We are very glad that the additional statistical analysis has fully addressed your concerns. We sincerely appreciate your constructive comments throughout the review process, which have helped us improve the paper.

---

### Official Review · Reviewer_1KBa · 2026-03-12

**Soundness:** 3
**Presentation:** 3
**Significance:** 2
**Originality:** 3
**Overall Recommendation:** 3
**Confidence:** 4

**Summary:**

This paper addresses the challenge of evaluating LLM as decision modules for long-horizon adversarial scenarios, and designs a diagnostic real-time strategy (RTS) benchmark based on Red Alert. The effectiveness of this benchmark is verified through multi-track experiments. Ultimately, the paper proposes future research directions for LLM agents and analyzes both the positive and negative impacts of the study.

**Compliance With Llm Reviewing Policy:**

Affirmed.

**Final Justification:**

I will raise the soundness score and keep my overall score.

**Key Questions For Authors:**

1. Questions regarding the design of evaluation metrics
Have the authors conducted a correlation analysis between process-level metrics (MC, MU, KLR, etc.) and long-horizon RTS victory outcomes? Can you specify the critical thresholds or optimal ranges for each metric, as well as the coupling mechanisms between scouting coverage, economic metrics, and combat loss ratio? Meanwhile, why was the temporal decomposition of all metrics across different phases (early-game development / mid-game confrontation / late-game decisive battle) not performed? Have you considered supplementing such analysis to locate the specific failure nodes of LLMs in different stages of long-horizon decision-making?

2. Questions regarding the evaluation method for human steerability
In the assessment of human instruction following in Track C, why did the authors only adopt standardized, pre-triggered instruction forms instead of introducing ambiguous instructions and dynamically revised instructions common in real human interactions (e.g., adjusting tactics midway or supplementing instruction details)? Additionally, have you considered designing test scenarios with erroneous instructions to evaluate LLMs’ ability to identify, reject, or correct unreasonable instructions, thereby making the evaluation more consistent with real human-AI interaction scenarios?

3. Questions regarding the proposed future research directions
For the specific problems identified in the experiments, such as LLMs’ inadequate ability to coordinate heterogeneous units, poor stability in tactical execution of some models, and the easy degradation of scouting/economic strategies in long-horizon decision-making, do the authors have more targeted technical improvement ideas? For instance, in terms of fine-tuning paradigms, prompt engineering design, and model memory mechanism optimization, can you provide specific research directions or experimental schemes instead of merely proposing generalized research goals?

**Limitations:**

yes

**Strengths And Weaknesses:**

Strengths
1. The benchmark framework features a hierarchically decoupled and standardized design, ensuring evaluation fairness
The LLM’s output of high-level macro/tactical intents is decoupled from low-level command execution. LLMs only generate low-frequency, budgeted intents, with low-level operations executed by a deterministic executor, which eliminates the interference of execution skills such as actions per minute (APM).

2. An improved Elo rating model is adopted for evaluating competitive strength, enhancing the scientific rigor of rankings
Elo-style ratings are calculated based on the Davidson extension of the Bradley–Terry paired-comparison model, which is compatible with draw outcomes and supports incomplete match sequences.

Weaknesses
1. The design of evaluation metrics lacks validation and temporal dimension analysis
First, the process-level metrics introduced in the paper, such as MC (per-minute economic income), MU (resource utilization rate), and KLR (combat loss ratio), have not undergone theoretical and empirical validation. The paper neither specifies the correlation thresholds between these metrics and long-horizon victory nor analyzes the coupling relationships among them (e.g., the mutual influence between scouting coverage and KLR). Second, all metrics are calculated as full-match averages without temporal analysis across different phases (early-game development / mid-game confrontation / late-game decisive battle). This fails to capture the changes in LLM capabilities and failure nodes across different stages of long-horizon decision-making, thus reducing the diagnostic value of the metrics.

2. The evaluation method for human steerability is overly rigid
The assessment of human instruction following in Track C only uses standardized instructions triggered at predefined times or states, without considering the ambiguity, dynamicity, and revisability of instructions in real human interactions (e.g., humans adjusting tactical instructions midway or describing strategies with ambiguous language). Nor does it evaluate LLMs’ ability to identify and correct erroneous instructions, leading to evaluation results for this dimension that fail to reflect LLMs’ actual steerability in real human-AI interactions.

3. The proposed future research directions lack specificity
The paper puts forward overly generalized future research directions such as "developing interaction-aware and adaptive LLM agents" and "strengthening persistent information state management", without proposing targeted solutions for the specific problems identified in the experiments. For example, it does not provide concrete methods for fine-tuning, prompt engineering, or architectural improvement to address LLMs’ inadequate ability to coordinate heterogeneous units, nor does it propose specific strategies for optimizing memory mechanisms to tackle the issue of decision degradation in LLMs during long-horizon tasks.

---

> ### Author Rebuttal · Authors · 2026-03-31
>
> ### W1 & Q1
>
> We conducted correlation and multivariate analyses between process-level metrics and win rate. The main finding is that no single metric is universally decisive, so we do not claim a fixed threshold or optimal range that transfers across settings. In Track A, MU has the highest univariate correlation with win rate (r = 0.95), but its independent contribution becomes very small in multivariate regression (β = 0.012), because it is strongly coupled with KLR (r = 0.73) and Scouting (r = 0.75). Thus, MU is better viewed as a proxy for a broader capability cluster combining combat efficiency and information advantage, rather than as an independent driver.
>
> The joint effect of metrics is much more informative than any single one. The four-metric model achieves R² = 0.972 in Track A and 0.994 in Track B, clearly exceeding the best single metric. KLR and Scouting are consistently helpful across tracks, while MU is more predictive in Track A and MC is relatively weak. In Track B, Scouting becomes the strongest factor (β = 0.238) and MU turns negative (β = -0.152), suggesting that against adaptive opponents, information advantage matters more than fine-grained resource optimization. Overall, victory arises from the interaction of combat, economy, and scouting rather than from optimizing one metric alone. We will add this correlation/coupling analysis in the revision.
>
> #### Correlation between Metrics and Win Rate
>
> | Metric   | Description | Track A | Track B|
> |----------|-------------------------------------|-----------------|--------------------|
> | KLR      | Kill-Loss Ratio                   | r = 0.78*       | r = 0.79*          |
> | MC       | Mineral Income per Minute           | r = 0.54        | r = -0.33          |
> | MU       | Military Utility (Total Expenditure)| r = 0.95        | r = 0.67           |
> | Scouting | Exploration Score                   | r = 0.79*       | r = 0.86*          |
>
> #### Multi-metric Regression Analysis
>
> | Track   | R²    | Adjusted R² | Key Drivers (Standardized β)                          |
> |---------|-------|-------------|--------------------------------------------------------|
> | Track A | 0.972 | 0.944       | KLR (0.097) > MC (0.05) > Scouting (0.046) > MU (0.012) |
> | Track B | 0.994 | 0.988       | Scouting (0.238) > MU (-0.152) > KLR (0.128) > MC (0.043) |
>
> We did not perform early-/mid-/late-game decomposition because phase boundaries are not well-defined in our setting. Both sides start with 20k credits and can allocate resources freely from the beginning, so scouting, expansion, and combat may start and interleave very differently across matches. Under this free-form annihilation setup, a fixed temporal split would be arbitrary and would not map reliably to comparable strategic stages.
>
> For this reason, capability-based analysis is more appropriate than phase-based analysis here. Our framework already localizes failure modes through scouting, combat efficiency, resource use, and command following, which are more stable and directly tied to decision quality. We will clarify this design choice in the revision.
>
> ### W2 & Q2
>
> Thank you for this profound insight. Your point accurately identifies the limitations in our current Track C design. Our initial design prioritized experimental control and reproducibility. Adopting standardized pre-triggered instructions facilitates model comparison under controlled conditions. However, we agree that this design does not fully capture the complex challenges of steerability in the real world.
>
> In the revised version, we will explicitly acknowledge this limitation and propose an extended evaluation framework in the appendix. This framework will include **ambiguous instructions requiring model clarification**, **dynamically revised instructions during gameplay**, and **deliberately erroneous instructions to test model robustness**. Although time constraints may prevent us from fully implementing this extended evaluation, we will provide detailed protocol specifications for future research reference.
>
> ### W3 & Q3
>
> As discussed in our conclusion and experiments, our results point to two concrete directions for future work. First, the findings from Track A and Track B suggest that future LLM decision systems should move beyond average-case performance toward **opponent-aware adaptive decision-making**. The large variance across opponent types, together with the fact that GPT-4o performs well against scripted rule-based opponents but poorly against Gemini-2.5-Flash, indicates that strong performance against fixed opponents does not transfer automatically to adaptive interactions. Second, the Track C results suggest a shift from **static prompt guidance to human-in-the-loop hierarchical control**. Since instruction-following quality declines as task complexity increases, future systems should not rely solely on prompting, but instead support continuous human guidance. Our benchmark highlights the importance of these two directions.

---

> > ### Author Rebuttal · Reviewer_1KBa · 2026-04-03
> >
> > Thanks for your response, and I will raise the soundness score and keep my overall score.

---

> > > ### Author Response · Authors · 2026-04-07
> > >
> > > Thank you for your thoughtful follow-up and for your continued engagement with our paper. We sincerely appreciate your careful reading of our response, and we are grateful for your decision to raise the soundness score.
> > >
> > > We will further clarify the scope of the paper, make the limitations more explicit, and improve the discussion of these points in the final version. Thank you again for your constructive feedback.

---

### Official Review · Reviewer_6AdS · 2026-03-13

**Soundness:** 3
**Presentation:** 3
**Significance:** 2
**Originality:** 2
**Overall Recommendation:** 3
**Confidence:** 4

**Summary:**

Using an existing open source agent harness for the RTS game red alert, the authors conduct an evaluation into LLM behaviour and long horizon performance (vs classical AI, LLMs and whether LLMs can follow human prompted behaviours).

**Compliance With Llm Reviewing Policy:**

Affirmed.

**Key Questions For Authors:**

please see strengths/weaknesses

**Limitations:**

please see strengths/weaknesses

**Strengths And Weaknesses:**

- APM is not defined.
-Is optimal gameplay in the training data? It would be useful to briefly discuss whether RTS gameplay data (e. strategy guides, replays, OpenRA discussions ..) might appear in training corpora and whether this could influence results.
-What are the main findings? My understanding is that the main takeaway is that models perform differently across auxiliary measures (economy, scouting, combat efficiency), and that excelling at one auxiliary metric does not necessarily correspond to winning. If that is the case, it might be helpful to state this more clearly and discuss which metrics, if any, actually correlate with win rate.
-For C1–3 how robust are these metrics? It is not clear to me how sensitive these metrics are to parameter choices or implementation details. Some discussion of robustness or validation of these measures would strengthen the evaluation.
-Good repeats. Five runs per match is appreciated, although given the variability of RTS games this may still be a fairly small sample size.
-Related to this, it would be helpful to see some discussion of statistical stability of the Elo-style rankings given the small number of matches.
-4o already has a high win rate against classical AI. To what extent is this benchmark already saturated once models reach a certain level of competence?
- Logs/interaction transparency. It would be helpful if the authors released logs of model–environment interaction (along with the existing screenshots which are great). In particular, it is not clear to what extent the benchmark measures strategic gameplay versus the model’s ability to correctly call the game-specific tooling / API.
- games are terminated after 30 mins. Having never played the game, it is tricky for me to gauge how long horizon this is. How many turns, how many tokens etc. I watched some gameplay on youtube and this helped, but I am still not confident the scale of the task.
-Finally, the paper states that no experience or memory is carried across games. This simplifies evaluation but also limits the “long-horizon agent” framing somewhat, since models are not adapting or learning across matches.

---

> ### Author Rebuttal · Authors · 2026-03-31
>
> Thank you for your thoughtful comments. We address each question in turn.
>
> ### W1.1 APM
>
> We will define APM explicitly. Our benchmark removes traditional low-level APM effects through batched high-level decisions and a unified deterministic executor: a model outputs an intent batch, execution finishes, and only then does the next round begin. Models decide every **47-64 seconds (0.93-1.27/min)**, so a 30-minute game contains about **28-38 strategic decision rounds**. **Latency is treated as part of capability**, not controlled away.
>
> ### W1.2 Pretraining overlap
>
> We cannot fully rule out RTS-related public data in pretraining. However, all models are evaluated under the same interface, rules, map, and budget, and we observe **strong cross-track ranking reversals** and **model-specific strengths** that are difficult to explain by memorization alone. The benchmark’s core tasks demand sustained multi-step decision-making, partial observability handling, and abstract tactical grounding—capabilities far beyond simple recall of game knowledge. Even if general RTS prior exists in model training, the observed **cross-model, cross-track performance disparities remain valid and informative** for diagnosing LLM long-horizon decision-making competencies.
>
> ### W1.3 Auxiliary measures
>
> We will state this more clearly. Auxiliary metrics do correlate with win rate, but their usefulness is context-dependent: KLR and scouting are consistently helpful, MU is strong in Track A but weaker in Track B, and MC is relatively weak. When considered jointly, however, the **four metrics explain much more of the win-rate variation than any single metric alone**, showing that victory comes from coordinated combat, economy, and information use.
>
> ### W1.4 Robustness of C1-C3
>
> We performed sensitivity analyses for all C1-C3 metrics using Kendall’s tau over parameter settings. The values are **0.819 for C1, 0.800 for C2, and 0.828 for C3**, indicating that **model rankings remain stable across reasonable configurations** rather than depending on one specific implementation.
>
> ### W1.5 Repeat
>
> We agree that five runs per matchup may appear limited in a stochastic RTS setting. To test this, we repeated the Rush AI evaluation 15 times and analyzed win rate, combat efficiency, macro economy, and map exploration. The **15-run results recover the same overall ordering** (GPT-4o > Gemini-2.5-Flash > Qwen-3-Max > DeepSeek-V3.2) and the same relative strengths, suggesting that **five runs are sufficient to recover stable comparative trends** in our setup. We will clarify this motivation and conclusion in the revision.
>
> ### W1.6 Elo-style ranking
>
> We quantified ranking stability with a pair-stratified bootstrap over Track B (1000 replicates). The **top-two ordering is highly stable**: Gemini-2.5-Flash ranks first in **99.1%** of replicates and GPT-4o ranks second in **99.1%**, with **score-difference confidence intervals that do not cross zero**. The Qwen-DeepSeek gap is less certain, which is expected because their direct matchup produced only draws. We will state the ranking conclusions at this confidence level.
>
> ### W1.7 Saturation of benchmark
>
> Although GPT-4o performs strongly in Track A, Track A is only one of three complementary tracks. In Track B, **meaningful matchup asymmetries remain**, and in Track C, **substantial command-following gaps remain across models**. Thus, the benchmark still has **clear discriminative power**. We have added a note in Section 4.2 to clarify that **Track A alone does not define overall capability**, and that the full benchmark suite provides a more nuanced assessment.
>
> ### W2. Release logs and API
>
> We agree that interaction transparency is important, and we will release logs as supplementary material, including environment observations, model outputs, and executed actions. In our retrospective check of past experiments, model outputs were able to invoke the game API correctly, suggesting that the benchmark primarily reflects **decision quality rather than failures in API calling**.
>
> ### W3.1 30-minute match
>
> We chose a 30-minute match because it requires sustained multi-stage planning rather than short-term reactions alone. As noted in Point 1, this horizon includes repeated high-level decisions on building, production, expansion, scouting, and attacking across about **28-38 rounds**.
>
> ### W3.2 Cross-game learning
>
> We agree that this benchmark does not evaluate cross-game adaptation or persistent memory. We focus on within-game long-horizon planning under partial observability and adaptive opponents, while keeping evaluation controlled and directly comparable across models. We will revise the framing to state this scope more precisely and position cross-game memory and adaptation as future work.
>
> **The additional analyses supporting our responses have been added to the Appendix.**

---

> > ### Author Rebuttal · Reviewer_6AdS · 2026-04-04
> >
> > Thank you for the detailed comments and additional experimentation.
> >
> > W1.1 APM: Ahhh actions per minute! Sorry, I don't quite follow your response, speed of response is or isn't a factor that affects benchmark performance? you say that the next round begins only after the model's actions have registered BUT then say "latency is treated as part of capability". Clarification here would be helpful.

---

> > > ### Author Response · Authors · 2026-04-07
> > >
> > > We thank the reviewer for this insightful clarification request regarding the APM metric and its interplay with response latency.
> > >
> > > To address your concern, we first clarify the distinction between low-level operational speed and high-level strategic reasoning in our benchmark. In standard RTS evaluations, APM (Actions Per Minute) typically quantifies the frequency of low-level, micro-management commands (e.g., unit control commands). This metric primarily reflects execution efficiency rather than high-level strategic planning ability.
> > >
> > > In our design, we explicitly decouple these two aspects to ensure fairness and focus:
> > >
> > > - **Batch Processing Mechanism:**
> > >  LLMs generate batched, multi-action intents at each decision step, with a strict upper bound on the number of actions per batch (the model freely chooses the count within this limit). All in-game operations are executed by a unified, deterministic executor. The next LLM decision round only initiates after all batched actions from the current round are fully completed. This mechanism strictly isolates the model's long-horizon decision-making and strategic planning from low-level operational efficiency, ensuring our evaluation core measures reasoning depth, not micro-operation speed.
> > >
> > > - **Impact of Response Latency:**
> > >  We further clarify that response latency—the time elapsed from receiving the prompt to generating the response (i.e., the interval between the completion of the last action and the start of the next decision round)—directly impacts decision-making efficiency. **Longer response times create gaps where no actions are executed, which disrupt the model's ability to dynamically respond to the evolving game state and can significantly alter match outcomes.**
> > >
> > > In our study, these variations in response latency are not controlled away; instead, they are treated as an intrinsic and evaluated characteristic of the model's long-horizon decision-making capabilities. This is because latency inherently represents the real-world trade-off between reasoning depth and decision speed in practical deployment scenarios. By retaining these differences, our benchmark provides a more realistic assessment of how LLM-based agents balance deliberation time and action efficiency in complex, real-time environments.
> > >
> > > Accordingly, we have revised the relevant confusing description in the original text from "This design preserves the long-horizon adversarial structure while decoupling high-level decision making from APM-like execution skill, enabling fairer comparisons and clearer diagnosis" to
> > > "This design preserves the long-horizon adversarial structure while decoupling high-level decision making from low-level execution efficiency, enabling fairer comparisons and clearer diagnosis."
> > >
> > > We sincerely appreciate your thoughtful and constructive feedback throughout the review process. If you feel that our rebuttal has addressed your concerns, we would be very grateful if you would consider taking this into account in your final evaluation.

---

### Official Review · Reviewer_P9kU · 2026-03-18

**Soundness:** 3
**Presentation:** 3
**Significance:** 2
**Originality:** 4
**Overall Recommendation:** 4
**Confidence:** 4

**Summary:**

In this paper, the authors propose a benchmark to evaluate the ability of LLMs in long-horizon decision-making, in contrast to existing static or short and medium-horizon benchmarks. Specifically, they use the game Red Alert RTS to assess different models across three tracks: playing against rule-based opponents, competing against other LLMs (cross-play), and evaluating the ability of the model to follow human commands (tactical).

Using a range of metrics, the authors show that LLM performance depends strongly on the evaluation setting. No single model consistently outperforms others across rule-based opponents, cross-play with other LLMs, and tactical benchmarks. The paper highlights that relying solely on average performance obscures important failure modes.

**Compliance With Llm Reviewing Policy:**

Affirmed.

**Final Justification:**

The authors have addressed my different concerns. I have increased my significance score but will maintain my overall score.

**Key Questions For Authors:**

1. In track A, it seems that across all models, draw is the most frequent outcome. Could you perhaps elaborate on this? Give the intuition on why this happen?
2. How does this work position itself to pure-reinforcement learning methods and more broadly generalize to the original objectif of long-horizon evaluation setting? Are there other relevant baselines (e.g., RL agents or heuristic planners).
3. How sensitive are the results to prompting choices? Did you observe significant variation in performance across different prompts or prompt formulations, especially against different rule-based opponents in Track A?
4. The objective of Track 3 is not clear to me. Against which type of opponent is the evaluation of Track 3 conducted? Also, do the results correlate with the chance of winning? I imagine a scenario where a low score on following human command could be explained because the LLM judged that it would reduce its chance of winning the game.

**Limitations:**

Yes

**Strengths And Weaknesses:**

**Strengths**

- The paper evaluates long-term decision-making capabilities of LLMs, which remain under-explored, as most existing benchmarks focus on static or short-horizon tasks
- The authors consider both open-weight and proprietary LLMs
- The paper is well structured
- The benchmark is well designed, with three complementary tracks (rule-based opponents, cross-play, human command following) and well-motivated and well-defined metrics that provide insight into model performance across different dimensions.
- The analysis against different styles of rule-based opponents in Track A, combined with process-level metrics (*scouting*, *economy*, and *combat exchange*), offers a detailed understanding of each model' behavior depending on the opponent style.
- The track C is a really interesting evaluation and the different scenarios are well explained and supported by visual examples in the appendix.
- The figures are easy to read and understand.


**Weaknesses**

- The experimental setup, particularly the Red Alert RTS environment, is insufficiently explained. It is unclear how a game is won, lost, or drawn, which makes interpreting results, especially in Track A, more difficult given the high number of draws. In general, I would recommend adding one paragraph on how explaining the game work, if there are multiple ways to win a game, describe the strategy of each of the rule-based opponent, etc.

- It is unclear how well the findings generalize to other long-horizon planning settings.

- The sensitivity of model performance to prompting is not discussed.

- Page 3, Section 3.2 (Combat exchange): the definitions of *LossCost_self* and *LossCost_enemy* are unclear and require clarification.

- Page 5, Section 4: the meaning of “default decoding” for each model is unclear.

- For Tracks A and B, the number of matchups is limited to five and to 3 for Track C, which may introduce variance and reduce confidence in the reported values.

- Figure 2 appears redundant with the last four columns of Table 1.

---

> ### Author Rebuttal · Authors · 2026-03-31
>
> ### W1. Red Alert RTS environment explanation & Q1. Why draw is the most frequent outcome in track A?
>
> Draws are frequent in Track A not because the 30-minute limit is too short, but because some matchups are intrinsically low-aggression and may remain unresolved even with a longer horizon. Decisively ending games usually finish within about 20 minutes, whereas non-aggressive matchups can otherwise continue for a very long time without reaching a natural termination. A player wins by eliminating all enemy buildings and combat units within the time limit, and loses once all of its own combat units are eliminated. If neither side is defeated within 30 minutes, the match is recorded as a draw.
>
> - **Turtle AI**: focuses on defense and slow buildup.
> - **Normal AI**: follows a balanced strategy.
> - **Rush AI**: emphasizes early attacks.
> - **Naval AI**: focuses on naval control and water-based attacks.
>
> ### W2. Generalization to other long-horizon planning scenarios is unclear
>
> We do not claim that Red Alert directly generalizes to all long-horizon planning domains. Instead, we position it as a diagnostic benchmark for core properties that recur across many long-horizon settings, including partial observability, delayed consequences, adaptive opponents, and sustained responsiveness to human interventions. Our motivation is precisely that many existing interactive benchmarks remain focused on shorter-horizon or non-adversarial tasks, making these failure modes difficult to observe.
>
> ### W3 & Q3. Sensitivity to prompt selection
>
> To assess this effect, we compared two prompt variants: a standard version (**our original prompt**) with full state descriptions and strategic guidance, and a simplified version with only the JSON constraint and a minimal task description.
>
> The simplified version leads to a clear drop in win rate. Without explicit subgoal decomposition and reasoning anchors, the model becomes less stable in balancing economic management and military actions. This suggests that, in complex RTS tasks, current LLMs still benefit from structured domain guidance to maintain consistent long-horizon decision-making.
>
> In our work, all main experiments use the empirically more stable standard prompt, so the reported results primarily reflect the method itself rather than prompt engineering effects. The simplified version is included only as an appendix ablation and does not affect our main conclusions.
>
> | Opponent | Standard W/D/L | Simplified W/D/L |
> |:---------|:--------------:|:----------------:|
> | **Overall** | 0.15 / 0.85 / 0 | 0 / 0.59 / 0.41 |
> | **Naval AI** | 0 / 1 / 0 | 0 / 1 / 0 |
> | **Normal AI** | 0.4 / 0.6 / 0 | 0 / 0.67 / 0.33 |
> | **Rush AI** | 0.2 / 0.8 / 0 | 0 / 0 / 1 |
> | **Turtle AI** | 0 / 1 / 0 | 0 / 0.67 / 0.33 |
>
> `W/D/L` denotes win, draw, and loss rate, respectively.
>
> ### W4. Definitions of LossCost_self and LossCost_enemy are unclear and require clarification
>
> - **LossCost_enemy**: total cost of enemy units and buildings destroyed during the match.
> - **LossCost_self**: total cost of our own units and buildings lost during the match.
>
> For example, a tank may cost from 300 to 1000.
>
> ### W5. Meaning of “default decoding”
>
> For transparency and reproducibility, all models use their **official native API default decoding settings**.
>
> ### W6. The number of matchups may introduce variance and reduce confidence in the reported values.
>
> We agree that five runs per matchup may appear limited in a stochastic RTS setting. To test this, we repeated the Rush AI evaluation 15 times and analyzed win rate, combat efficiency, macro economy, and map exploration. The **15-run results recover the same overall ordering** (GPT-4o > Gemini-2.5-Flash > Qwen-3-Max > DeepSeek-V3.2) and the same relative strengths, suggesting that **five runs are sufficient to recover stable comparative trends** in our setup. We will clarify this motivation and conclusion in the revision. We are extending Track C to five runs and will report the updated results in the revision.
>
> ### W7. Redundancy between Figure 2 and Table 1
>
> We agree with the reviewer. In the revision, we will streamline the presentation.
>
> ### Q2. How does this work position itself to RL agents or heuristic planners？
>
> Our benchmark is designed primarily as a diagnostic testbed for LLM long-horizon decision-making. While pre-trained RL agents and heuristic planners are valuable as potential supplementary baselines, our current goal is to provide a standardized comparison within the LLM family to better characterize their inherent long-horizon planning capabilities.
>
> ### Q4. The objective of Track C is unclear.
>
> The objective of Track C is to evaluate controllability under human intent, which asks a different question: can an agent faithfully ground high-level human tactical instructions into coordinated multi-unit behavior over time, rather than merely achieving a good final outcome?
>
> The opponent in Track C is manually designed and uses automatic firing.

---

> > ### Author Rebuttal · Reviewer_P9kU · 2026-04-02
> >
> > The authors have addressed all of my concerns. My only recommendation is to include a brief description of the game’s goal. Adding one or two sentences would help the reader better grasp the objectives of the benchmark and clarify where this work fits within the RL space.

---

> > > ### Author Response · Authors · 2026-04-07
> > >
> > > Thank you for your positive feedback. In the revised version, we will add more descriptions to enhance clarity.
> > >
> > > In our Red Alert environment, both the red and blue sides start with an initial base, and the objective is to destroy all enemy combat units and buildings, including the starting base.
> > >
> > > The benchmark is designed to evaluate whether an LLM can maintain coherent long-horizon decision-making in a real-time adversarial environment: specifically, whether it can repeatedly make and update strategic choices on resource allocation, production, scouting, attack, and defense over the course of a full match, instead of solving a short-horizon or single-step task. The goal is therefore not to measure low-level control alone, but to test sustained strategic reasoning under delayed consequences, partial information, and continuous opponent interaction.
> > >
> > > Our Red Alert setting is well matched to this objective because it is a real-time strategy game with a dynamic adversarial state space, where decisions made at one moment constrain and shape future options. In our setup, both players begin from an initial base and must ultimately destroy the opponent’s units and structures. During the match, the agent must repeatedly balance economy, army production, map exploration, and combat response while reacting to the opponent’s changing behavior and unexpected battlefield events. For these reasons, the benchmark is suitable for evaluating whether LLMs can sustain long-term strategic coherence and adapt their plans throughout an evolving match.
> > >
> > > We will also make the connection to the RL space more explicit. Although our focus is on evaluating LLM-based decision modules rather than training RL agents, the benchmark directly instantiates core sequential decision-making challenges central to RL, including long horizons, partial observability, delayed consequences, and adversarial interaction. In principle, trained RL agents could also be tested in this benchmark, but this would place relatively high demands on the underlying RL algorithm, because the decision interface entails a large decision space that combines production choices (e.g., selecting which unit to build) with coordinated control over one or multiple units toward target locations. We believe this addition will make the benchmark’s objective and positioning clearer to readers.
> > >
> > > We sincerely appreciate your constructive feedback throughout the review process. Since you indicated that your concerns have been addressed, we would be very grateful if you would consider reflecting this in your final score.

---

### Decision · Program_Chairs · 2026-04-30

**Decision:**

Accept (regular)

**Comment:**

Reviewers agree this benchmark using the Red Alert RTS game to evaluate the ability of LLMs in long-horizon decision-making is generally well-designed and well-motivated, with potential to provide insight into model performance. Reviewers were concerned by several areas that lacked clarity and full empirical investigation, such as how the game is won/lost/drawn, whether findings generalize, prompting sensitivity, response latency, metric empirical validation, future directions, and robustness over 5 runs. These were largely addressed in the author responses. At least one concern about using standardized instructions vs real human interactions was not fully resolved in the author response, though authors committed to discussing this limitation further in the paper. Authors should be sure to take reviewer feedback into account in their revision.